# Position of UNC-13 in the active zone regulates synaptic vesicle release probability and release kinetics

**Keming Zhou[1], Tamara M Stawicki[1,2†], Alexandr Goncharov[1,3], Yishi Jin[1,3*]**

[1]Section of Neurobiology, Division of Biological Sciences, University of California, San Diego, La Jolla, United States; [2]Neurosciences Graduate Program, University of California, San Diego, La Jolla, United States; [3]Howard Hughes Medical Institute, University of California, San Diego, La Jolla, United States

**Abstract** The presynaptic active zone proteins UNC-13/Munc13s are essential for synaptic vesicle (SV) exocytosis by directly interacting with SV fusion apparatus. An open question is how their association with active zones, hence their position to $Ca^{2+}$ entry sites, regulates SV release. The N-termini of major UNC-13/Munc13 isoforms contain a non-calcium binding $C_2A$ domain that mediates protein homo- or hetero-meric interactions. Here, we show that the $C_2A$ domain of *Caenorhabditis elegans* UNC-13 regulates release probability of evoked release and its precise active zone localization. Kinetics analysis of SV release supports that the proximity of UNC-13 to $Ca^{2+}$ entry sites, mediated by the $C_2A$-domain containing N-terminus, is critical for accelerating neurotransmitter release. Additionally, the $C_2A$ domain is specifically required for spontaneous release. These data reveal multiple roles of UNC-13 $C_2A$ domain, and suggest that spontaneous release and the fast phase of evoked release may involve a common pool of SVs at the active zone.

**\*For correspondence:** yijin@ucsd.edu

**†Present address:** Department of Biological Structure, University of Washington, Seattle, United States

**Competing interests:** The authors declare that no competing interests exist.

## Introduction

The molecular mechanism of how $Ca^{2+}$ influx accelerates synaptic vesicle (SV) release remains at the heart of understanding synapse action in the normal brain and under disease conditions (*Jahn and Fasshauer, 2012*). In most synapses, SV release evoked by $Ca^{2+}$ influx consists of a fast synchronous phase that occurs on a millisecond timescale and a slow asynchronous phase that occurs with some delay and persists for tens or hundreds of milliseconds. Additionally, all synapses manifest spontaneous release, corresponding to stochastic fusion of individual SVs. The key proteins mediating SV release include soluble Nethylmaleimide–sensitive factor attachment protein receptor (SNARE) proteins, Sec1/Munc18 (SM) proteins, UNC-13/Munc13s, synaptotagmins and complexins (*Wojcik and Brose, 2007*; *Südhof and Rothman, 2009*).

The presynaptic active zone is enriched with $Ca^{2+}$ channels (*Meinrenken et al., 2002*) and cytomatrix proteins that organize the action of presynaptic release through multi-domain protein interaction network (*Schoch and Gundelfinger, 2006*; *Südhof, 2012b*). The mechanisms for why some SVs release rapidly and others slowly in response to membrane depolarization have been primarily attributed to the heterogeneity in their intrinsic $Ca^{2+}$ sensitivities, such that a low-affinity $Ca^{2+}$ sensor promotes the fast phase, while a high-affinity $Ca^{2+}$ sensor supports the slow phase (*Südhof, 2012a*). Yet, numerous studies have suggested that the distance between SVs and $Ca^{2+}$ entry sites is also a critical determinant for release kinetics and release probability (*Neher and Sakaba, 2008*; *Hoppa et al., 2012*). For example, in the calyx of Held, a giant synapse in the brainstem, it was shown that SVs involved in the slow phase of evoked release triggered by prolonged depolarization are as sensitive to $Ca^{2+}$ as those in the fast phase, and can undergo rapid release upon uniform elevation of intracellular

**eLife digest** Neurons are connected to each other by junctions called synapses. When an electrical signal travelling along a neuron arrives at a synapse, it causes the release of bubble-like structures called synaptic vesicles that contain chemicals called neurotransmitters. When released by the vesicles these neurotransmitters bind to receptors on a second neuron and allow the signal to continue on its way through the nervous system.

The release of synaptic vesicles from the neuron depends largely on the number of calcium ions that enter this neuron via structures called ion channels, and also on the rate at which they enter. Vesicles are released in one of three ways: they can be released quickly (within a few milliseconds) in response to the influx of calcium ions; they can be released slowly (over a period of tens or hundreds of milliseconds) in response to the influx; or they can be released at random times that are not related to the influx.

It is known that the sensitivity of certain calcium sensors near the synapse influences the release of the vesicles. It had been thought that the distance between the "active zone" where the calcium ions enter the neuron and the region where the vesicles reside might also influence rate of release, but the molecular mechanism underlying this hypothesis is poorly understood.

Zhou et al. have now shed new light on this question by performing a series of experiments that involved manipulating a protein called UNC-13 – which is known to be involved in the release of vesicles – in neurons from *C. elegans*, a nematode worm. First it was shown that the precise position of UNC-13 in the active zone depended on a domain within the protein called the $C_2A$ domain. Next it was shown that the distance between the UNC-13 protein and the calcium ion channels strongly influences the quick mode of vesicle release. Finally, Zhou et al. showed that the $C_2A$ domain also had a significant influence on the spontaneous release of vesicles, which suggests that a common fleet of vesicles might be used for both the quick and the spontaneous modes of vesicle release. Zhou et al. also generated mutant worms that mimicked a neurological disease, epileptic seizure, and showed that eliminating the $C_2A$ domain can relieve some of the symptoms associated with the disease.

Many neurological diseases are caused by signals not being transmitted properly at synapses, so in addition to providing insights into the basic mechanism underlying synaptic action, these results could also assist with the development of new strategies for managing neurological diseases.

Ca$^{2+}$ (*Wadel et al., 2007*). A recent study also showed that presynaptic over-expression of an auxiliary α2δ subunit of voltage-gated calcium channels (VGCCs) led to a dramatic increase in release probability even though the total Ca$^{2+}$ influx was reduced. It was suggested that the α2δ subunit may promote a closer spatial correlation between sites of Ca$^{2+}$ influx and vesicle release (*Hoppa et al., 2012*). Nonetheless, the mechanism for distance mediated regulation of SV release remains poorly understood.

The UNC-13/Munc13 family of proteins are conserved core components of the presynaptic active zone, and are essential for both evoked and spontaneous SV release (*Augustin et al., 1999*; *Richmond et al., 1999*). UNC-13/Munc13 proteins contain multiple protein interaction domains, and have been linked to nearly all aspects of presynaptic release. Common to all protein isoforms are a diacylglycerol-binding $C_1$ domain followed by a MUN domain including the MHD (Munc13 homology domain) flanked by a $C_2B$ and a $C_2C$ domain (*Figure 1A*). The MUN domain is structurally similar to the vesicle tethering factors of the CATCHR (Complex Associated with Tethering Containing Helical rods) family (*Li et al., 2011*), and is necessary for vesicle priming (*Basu et al., 2005*; *Madison et al., 2005*; *Stevens et al., 2005*) through binding to SNARE and Munc18 (*Betz et al., 1997*; *Ma et al., 2011*). The N-terminal regions of UNC-13/Munc13 isoforms are divergent in amino acid sequences, and have been hypothesized to contribute to the distinct properties of SV exocytosis in different types of synapses (*Augustin et al., 2001*; *Rosenmund et al., 2002*). Of direct relevance, a non-calcium binding $C_2A$ domain resides at the N-terminus of the major isoforms, which include Munc13–1, ubMunc13–2 and *C. elegans* UNC-13 long isoform. This $C_2A$ domain can homodimerize (*Lu et al., 2006*), and heterodimerize with the zinc finger domain of the active zone protein RIM (*Betz et al., 2001*; *Dulubova et al., 2005*). RIM can tether presynaptic Ca$^{2+}$ channels to the active zone, and lack of RIM reduces

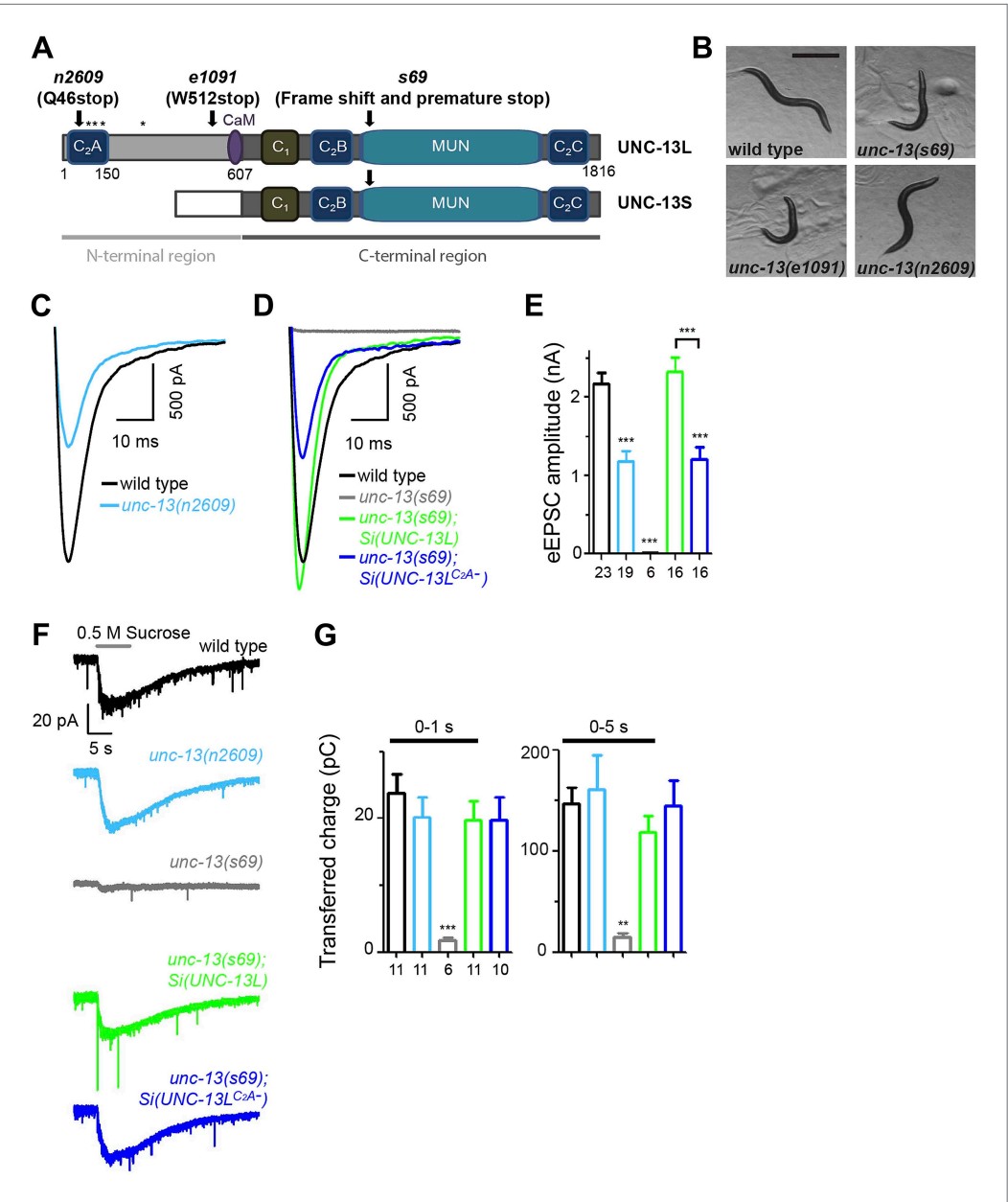

**Figure 1**. The $C_2A$ domain of UNC-13L regulates the release probability of evoked synaptic vesicle release.
(**A**) Illustration of UNC-13 long and short isoforms, and location of *unc-13* mutations. * marks possible initiation methionines downstream of *n2609* mutation. The purple domain is the calmodulin binding site (CaM). (**B**) Bright field images of adult animals from wild type, *unc-13(s69)*, *unc-13(e1091)* and *unc-13(n2609)*. Scale bar: 0.5 mm.
(**C** and **D**). Average recording traces of eEPSCs in animals of genotype indicated. (**E**). Summary of peak amplitudes of eEPSCs from genotypes shown in **C** and **D**. (**F** and **G**). Average recording traces (**F**) and summary of transferred charges (**G**) of 0.5 M hypertonic sucrose solution induced vesicle release in animals of genotype indicated. The number of animals analyzed is indicated for each genotype. Error bars in **E** and **G** indicate SEM. Statistics, one way ANOVA. ***p<0.001.

The following figure supplements are available for figure 1:

**Figure supplement 1**. Alignment of $C_2A$ domains among UNC-13/Munc13 isoforms.

**Figure supplement 2**. Transcripts of *unc-13(n2609)*.

*Figure 1. Continued*

**Figure supplement 3**. The effects of loss of the C$_2$A domain on locomotion speeds.

**Figure supplement 4**. Ratios of mean charge transfers during eEPSC and during sucrose application and the rescue effects of overexpression of UNC-13L and UNC-13L$^{C2A-}$ in *unc-13(s69)*.

Ca$^{2+}$ channel density and Ca$^{2+}$ influx at active zones (**Han et al., 2011**; **Kaeser et al., 2011**; **Muller et al., 2012**). Recently, an elegant study has demonstrated a mechanism in which homodimerization by the C$_2$A domain keeps ubMunc13–2 under priming-inhibitory state, while heteromeric binding between the C$_2$A domain of ubMunc13–2 and the zinc finger of RIM relieves the inhibition to promote SV priming (**Deng et al., 2011**). However, a direct test for C$_2$A domain's function in vivo is not yet shown.

In *C. elegans* the *unc-13* locus produces two main isoforms that differ at the N-terminal region (**Figure 1A**) (**Kohn et al., 2000**). The abundant long isoform UNC-13L closely resembles Munc13–1 and ubMunc13–2. Previous studies using genetic mutations that eliminate function of all isoforms or UNC-13L demonstrate an essential role of UNC-13L in neurotransmitter release (**Richmond et al., 1999**). A recent study reveals that UNC-13L is involved in both fast and slow release of SVs, while the short isoform UNC-13S is required for the slow release (**Hu et al., 2013**). Here, we identified a unique *unc-13* mutant that specifically deletes the C$_2$A domain of UNC-13L. Exploiting this mutant, we show that the C$_2$A domain regulates the release probability of SVs, likely through positioning UNC-13L to the active zone. The C$_2$A domain also has a specific role in spontaneous release. Loss of the C$_2$A domain of UNC-13L blocks the enhanced spontaneous release caused by loss of complexin. Furthermore, using the genetically encoded photosensitizer miniSOG (mini singlet oxygen generator), we find that acute ablation of the active-zone specific UNC-13L results in a strong inhibition of spontaneous release and of the fast phase of evoked release, while ablation of a non-active-zone variant of UNC-13L alters primarily the slow phase of evoked release. These observations support an idea that spontaneous release and the fast phase of evoked release may use a common pool of SVs. We also show that reducing SV release by eliminating the function of UNC-13L C$_2$A domain ameliorates behavioral deficits in a *C. elegans* model for epileptic seizure. Together, these data demonstrate that the distance between UNC-13/Munc13 to the Ca$^{2+}$ entry site plays a critical role in SV release probability and release kinetics.

## Results

### A phenotypically unique *unc-13* mutant lacking the C$_2$A domain

The *unc-13* locus contains 31 exons, and produces a major long isoform UNC-13L of 1816 amino acid residues and a short isoform UNC-13S that lacks the N-terminal 607 amino acid residues of UNC-13L and has a different N-terminal domain, through the use of different promoters and alternative splicing (**Figure 1A**, **Figure 1—figure supplement 2**) (**Kohn et al., 2000**). We isolated the *unc-13(n2609)* mutation in a genetic screen for suppression of the convulsive behavior caused by a gain-of-function mutation in the neuronal acetylcholine receptor *acr-2* (**Jospin et al., 2009**) (see 'Materials and methods' and below). *unc-13(n2609)* changes glutamine 46 to a stop codon within the C$_2$A domain of UNC-13L (**Figure 1A**). The C$_2$A domain shares 50% identity with rat Munc13–1 and 49% identity with rat ubMunc-13–2, respectively (**Figure 1—figure supplement 1**). The key amino acids for homodimerization and for heterodimerization with RIM are conserved between *C. elegans* and mammals. Among previously reported mutations, *unc-13(s69)* is a null allele for the entire locus, and *unc-13(e1091)* is a null allele for the long isoform only (**Figure 1A**, **Supplementary file 1A**). Both *unc-13(s69)* and *unc-13(e1091)* mutants are severely paralyzed (**Kohn et al., 2000**) (**Figure 1B**). In contrast, *unc-13(n2609)* animals exhibited moderate slowing of locomotion (**Figure 1B**, **Figure 1—figure supplement 3**). *unc-13(n2609)/unc-13(s69)* or *unc-13(n2609)/unc-13(e1091)* animals showed more movement impairment than did *unc-13(n2609)* homozygous mutants (data not shown). These observations indicate that *unc-13(n2609)* is a recessive partial loss of function mutation.

The mild behavioral defects of *unc-13(n2609)* suggest that some UNC-13L proteins are produced in this mutant. Indeed, immunostaining using an antibody raised against amino acids 106–528 of the

UNC-13L N-terminus (*Kohn et al., 2000*) revealed strong staining in *unc-13(n2609)* (described later), while no immunostaining signal was detected in *unc-13(s69)* and *unc-13(e1091)* mutants (*Kohn et al., 2000*) (data not shown). We performed RT-PCR for *unc-13* transcripts, and confirmed that the *n2609* mutation did not alter *unc-13* splicing or cause skipping of the mutation-containing exon 3 (*Figure 1— figure supplement 2*). The Q46 to stop codon mutation was present in cDNA clones generated from the mutant strain. Thus, the observed expression of UNC-13L proteins in *unc-13(n2609)* mutant must be due to translation from an ATG codon(s) downstream of the amino acid Q46 (*Figure 1A*, *Figure 1— figure supplement 1*). To rule out the possibility that loss of additional N-terminal sequences in UNC-13L might contribute to the phenotypes in *unc-13(n2609)*, we next generated single-copy insertion transgenes (designated *Si*) expressing the full length UNC-13L or a mutant UNC-13L$^{C2A-}$ lacking only the $C_2A$ domain (deleting amino acids 1–152, *Figure 1—figure supplement 3*) driven by pan-neuronal promoter, using the transposon Mos-mediated insertion (MosSCI) technique (*Frøkjær-Jensen et al., 2008*). While *Si(UNC-13L)* transgene fully rescued the locomotion of *unc-13(s69)* null mutants, *Si(UNC-13L$^{C2A-}$)* rescued the movement deficits of *unc-13(s69)* to a level similar to *unc-13(n2609)* mutants (*Figure 1—figure supplement 3*). Thus, we conclude that the *unc-13(n2609)* mutant is specifically deficient in the UNC-13L $C_2A$ domain, and provides a genetic background to investigate the role of the $C_2A$ domain in a physiological setting.

## The $C_2A$ domain of UNC-13 regulates the release probability of synaptic vesicles

To define how lacking the $C_2A$ domain of UNC-13 affects synaptic physiology, we assessed synaptic transmission at the cholinergic neuromuscular junctions (NMJs) by electrophysiological recordings of muscle cells. SV release at these synapses occurs in a graded manner in response to membrane potential change (*Liu et al., 2009*). Under depolarizing condition, evoked excitatory post-synaptic currents (eEPSCs) represent simultaneous release of hundreds of SVs. *unc-13(s69)* null animals exhibit no eEPSCs (*Richmond et al., 1999*) (*Figure 1D*). In *unc-13(n2609)* mutants the amplitude of eEPSCs was reduced to 50% of that in wild type animals (*Figure 1C,E*). We also performed electrophysiological recording in the *unc-13(s69)* null animals expressing full-length UNC-13L or UNC-13L$^{C2A-}$ from the same integrated genomic locus. We observed a full rescue of the amplitude of eEPSC by *Si(UNC-13L)* expression in *unc-13(s69)* (*Figure 1D,E*). *Si(UNC-13L$^{C2A-}$); unc-13(s69)* animals showed significantly reduced amplitude of eEPSC, similar to *unc-13(n2609)*. To address that the $C_2A$ domain is directly responsible for the observed physiological defect, not secondary due to reduced protein levels, we overexpressed full-length UNC-13L and UNC-13L$^{C2A-}$ in *unc-13(s69)* mutants. While both transgenes rescued the paralysis of *unc-13(s69)*, NMJ recordings showed that simply elevating the levels of UNC-13L$^{C2A-}$ did not fully rescue the eEPSC amplitude, compared to overexpression of the full-length UNC-13L (*Figure 1—figure supplement 4B*). Thus, these analyses strongly support that the $C_2A$ domain is required for $Ca^{2+}$ influx evoked SV release.

The reduced presynaptic release in *unc-13(n2609)* could be due to defective priming of SVs or a weak response of SVs to $Ca^{2+}$ influx at the presynaptic terminal. A classic assay to analyze SV priming is by the application of hypertonic sucrose solution to induce vesicle exocytosis in a $Ca^{2+}$-independent manner, which is often used to assess readily releasable pool (RRP) (*Rosenmund and Stevens, 1996*). Previous reports have shown that SV priming under brief (1 s) sucrose application is almost abolished in *unc-13(s69)* mutants and severely inhibited in *unc-13(e1091)* mutants (*Richmond et al., 1999*; *Madison et al., 2005*). Here we applied a prolonged sucrose stimulation protocol to release the majority of primed vesicles. This protocol enabled us to assess the charge transfer with better time resolution at the first second and initial 5 s of sucrose application. Under this protocol, the charge transfer during the first second was 23.7 ± 2.9 pC in wild type animals and was 1.7 ± 0.5 pC in *unc-13(s69)* mutants (*Figure 1G*), comparable to previous reports using a brief sucrose stimulation (*Gracheva et al., 2006*; *McEwen et al., 2006*). Prolonged sucrose application did not induce further release in *unc-13(s69)*. Sucrose-induced charge transfers in the time windows of first one and 5 s were similar between wild type and *unc-13(n2609)* (*Figure 1F,G*). Both *Si(UNC-13L)* and *Si(UNC-13L$^{C2A-}$)* transgenes rescued SV priming in *unc-13(s69)* null mutants to the level of wild-type. These results indicate that SVs are fully competent for release in the absence of the $C_2A$ domain of UNC-13L.

The extended current evoked by sucrose stimulation under our protocol may reflect continuous release of refilled SVs to RRP (*Deng et al., 2011*; *Watanabe et al., 2013*). Because the preparation for *C. elegans* NMJ recording cannot endure multiple stimulations, it is not feasible to record reliable

responses for multiple stimulations to compare the charge transfers during eEPSC and sucrose application in the same animal. We therefore calculated the ratio of mean charge transfers during eEPSC and sucrose application for a given genotype. This ratio may not directly represent the release probability, but is positively correlated with release probability. In *unc-13(n2609)* mutants and *unc-13(s69); Si(UNC-13L^{C2A−})* transgenic animals, this ratio was severely reduced (*Figure 1—figure supplement 4A*). As *unc-13(n2609)* mutants display reduced evoked release but unaltered SV priming, we conclude that the $C_2A$ domain of UNC-13L regulates the release probability of SVs.

## The $C_2A$ domain of UNC-13 contributes to synaptic vesicle docking at the active zone

SV priming and release probability are generally correlated with the number of docked SVs (*Schikorski and Stevens, 2001*; *Holderith et al., 2012*). The requirement of UNC-13 for SV docking at the active zone has been revealed by ultrastructural analyses of synapses using high pressure freezing fixation (*Weimer et al., 2006*; *Hammarlund et al., 2007*). In *unc-13(s69)* and *unc-13(e1091)* mutants, fewer docked SVs are present within 231 nm from presynaptic dense projections, while slightly more SVs are accumulated at distal regions (>330 nm from presynaptic dense projections) (*Hammarlund et al., 2007*). To address if the $C_2A$ domain of UNC-13L influences docking of SVs at active zones, we examined the distribution of SVs using the high pressure freezing fixation protocol (*Weimer et al., 2006*; *Hammarlund et al., 2007*). Neuromuscular synapses in *unc-13(n2609)* showed normal ultrastructural organization (*Figure 2A*). Consistent with the normal SV priming in the *unc-13(n2609)* mutant, the number of total SVs and that of total docked SVs were similar to those in wild type animals (*Figure 2C*). However, fewer docked SVs were present in the central active zone (0–165 nm) and more docked SVs were present distally (>330 nm) (*Figure 2B,D*). Although this SV docking defect in *unc-13(n2609)* is less severe than those in *unc-13(s69)* and *unc-13(e1091)* mutants (*Hammarlund et al., 2007*), the mild reduction in the centrally docked SV in *unc-13(n2609)* may partially account for the reduced release probability.

## The $C_2A$ domain-containing N-terminus of UNC-13L determines its precise localization at the active zone

UNC-13/Munc13 proteins are core components of the presynaptic active zone and interact with multiple active zone proteins (*Kohn et al., 2000*; *Andrews-Zwilling et al., 2006*; *Südhof, 2012b*). The ultrastructural appearance of the presynaptic dense projection was grossly normal in *unc-13(n2609)*. To further test whether the recruitment of active zone proteins might be affected, we examined the localization of several active zone proteins, including the *C. elegans* RIM protein UNC-10 (*Koushika et al., 2001*), ELKS-1 (*Deken et al., 2005*), and the α1 subunit of presynaptic voltage-gated $Ca^{2+}$ channels (VGCCs) UNC-2 (*Jospin et al., 2007*). We found that the co-localization pattern of UNC-10 with ELKS-1 (*Figure 3A1–4*) and of UNC-10 with UNC-2::GFP (*Figure 3—figure supplement 1*) were indistinguishable between wild type and *unc-13(n2609)* animals. UNC-13L proteins, recognized by the antibodies against the N-terminus of UNC-13L, showed a punctate pattern in *unc-13(n2609)* mutants. However, UNC-13L puncta displayed significantly reduced co-localization with UNC-10/RIM (*Figure 3B1,B3*). The distance from an UNC-13L punctum to the nearest UNC-10/RIM punctum was significantly increased in *unc-13(n2609)*, compared to wild type (*Figure 3B2,B4*). We observed similarly altered UNC-13L and UNC-10/RIM colocalization in *unc-13(s69); Si(UNC-13L^{C2A−})* animals, comparing to *unc-13(s69); Si(UNC-13L)* (*Figure 3C1–4*). As UNC-10/RIM, ELKS-1 and UNC-2/VGCC are correctly recruited to the active zone in *unc-13(n2609)* mutants, these data indicate that lacking the $C_2A$ domain causes UNC-13L to be shifted away from the active zone where $Ca^{2+}$ entry sites reside.

The distribution pattern of UNC-13L proteins is grossly punctate in *unc-10/rim* mutants (*Koushika et al., 2001*) (*Figure 3—figure supplement 2*), suggesting that the presynaptic localization of UNC-13L is not solely dependent on UNC-10/RIM. Supporting this idea, GFP tagged $C_2A$ domain (N1–157) displayed a diffuse pattern throughout the axon (*Figure 4A*). It has been shown that the UNC-13S isoform, which has a different N-terminal domain, is diffusely localized throughout the cytoplasm (*Nurrish et al., 1999*). These observations imply that additional protein sequences of the N-terminus of UNC-13L may contribute to its active zone localization. Indeed, we found that the entire N-terminal region (N1–607) of UNC-13L tagged with GFP showed a punctate pattern similar to the full-length UNC-13L::GFP (*Figure 4A*). Conversely, removing the N-terminal domain, UNC-13L^{N−},

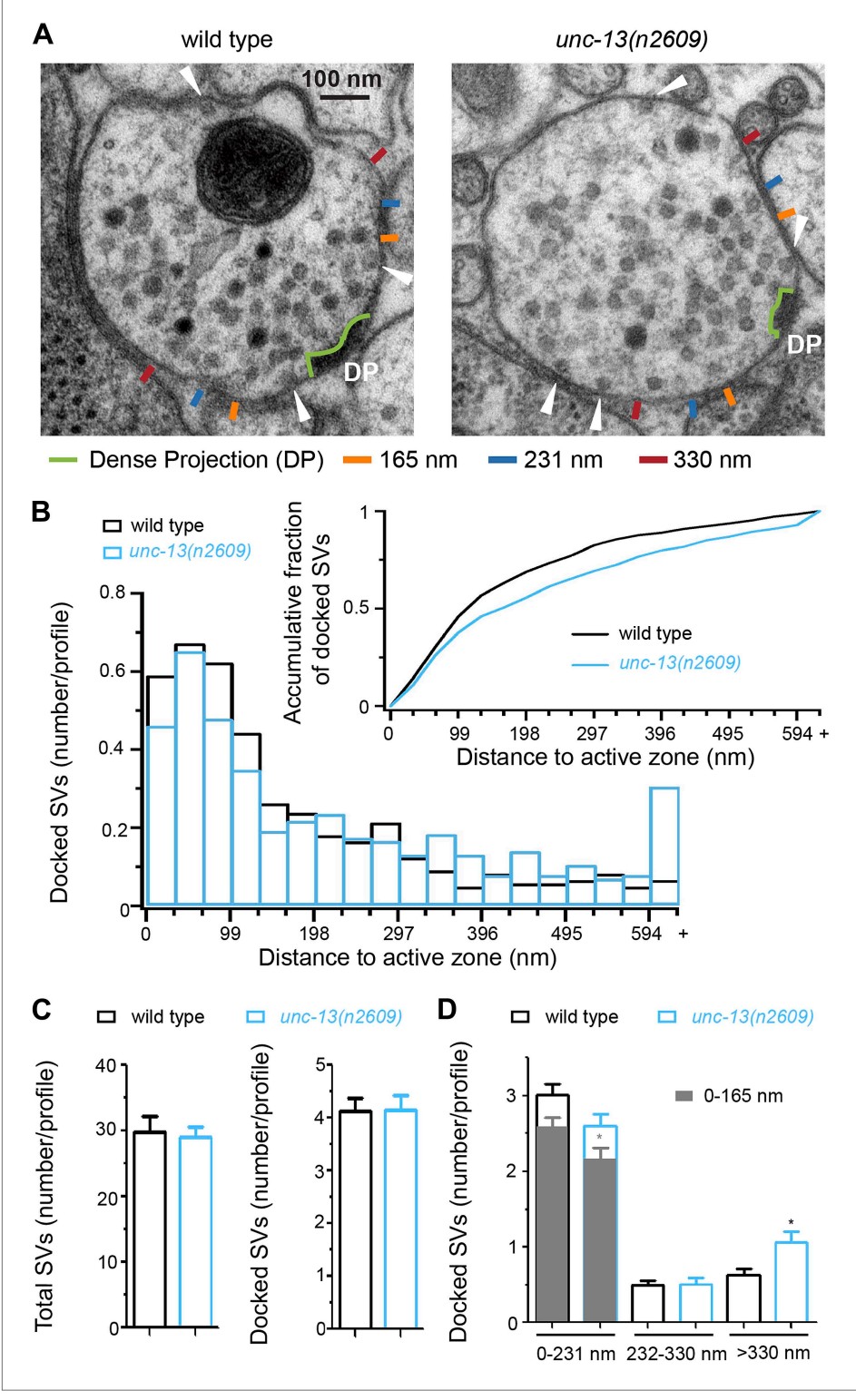

**Figure 2**. The C$_2$A domain of UNC-13L promotes the docking of synaptic vesicles at the active zone. (**A**) Ultrastructural organization of cholinergic presynaptic terminals in wild type and *unc-13(n2609)*. The dense projections were outlined by light green. The 165 nm, 231 nm and 330 nm regions along the plasma membrane from the edge of dense projection were marked by ticks with different colors. Docked synaptic vesicles are indicated by white arrowheads. (**B**) The histogram of docked vesicle number per profile located at different distances to the dense projection in wild type and *unc-13(n2609)*. Insert, Normalized accumulative distribution of *Figure 2. Continued on next page*

*Figure 2. Continued*

docked vesicles in wild type and *unc-13(n2609)*. (**C**) The average number of total synaptic vesicles (left) and docked synaptic vesicles (right) in single profiles of cholinergic synapse containing a dense projection are similar between wild type and *unc-13(n2609)*. (**D**). The average docked vesicle number per profile from each synapse in specific regions (<165 nm, <231 nm, 232–330 nm and >330 nm). Data were collected from one wild type animal (21 synapses, 122 profiles and 501 docked synaptic vesicles) and one *unc-13(n2609)* animal (25 synapses 115 profiles and 485 docked synaptic vesicles). Error bars indicate SEM in **C** and **D**. Statistics, two-tailed Student's *t* test. *p<0.05.

resulted in diffuse axonal localization. These data are consistent with the recent report (*Hu et al., 2013*) and show that both the $C_2A$ domain and additional N-terminal sequences of UNC-13L are responsible for its precise position in the presynaptic active zone.

## Precise localization of UNC-13L in the active zone is critical for fast kinetics of $Ca^{2+}$ triggered evoked release

It has been proposed that the distance of release competent SVs to sites of $Ca^{2+}$ influx strongly influences the release probability and kinetics of SV exocytosis (*Wadel et al., 2007*; *Hoppa et al., 2012*). Among presynaptic active zone proteins, UNC-13/Munc13 is unique in that it directly interacts with the SV fusion apparatus (*Betz et al., 1997*; *Ma et al., 2011*). To address the significance of UNC-13L localization in function, we first compared the activity of UNC-13L$^{N-}$, UNC-13L$^{C2A-}$, and full-length UNC-13L driven by the same pan-neuronal promoter from the same genomic insertion locus in rescuing the paralysis of *unc-13(s69)*. Quantitative analysis of locomotion speed showed a poor rescuing activity of *unc-13(s69)* paralysis by *Si(UNC-13L$^{N-}$)*, compared to *Si(UNC-13L)* and *Si(UNC-13L$^{C2A-}$)* transgenes (*Figure 4—figure supplement 1*). We next investigated how altered UNC-13L localization affected SV release kinetics. We analyzed eEPSCs with 90–10% decay time, which mainly reflects the duration of slow phase of evoked release, as well as the cumulative charge transfer of eEPSCs by fitting with a double-exponential function (see 'Materials and methods'). In wild type animals, eEPSCs lasted around 50 ms and decayed close to the baseline with 90–10% decay time being 18.56 ± 2.22 ms (*Figure 4B–C*). The charge transfer during eEPSC manifested two kinetic components with the time constants for the fast and slow components differing by a factor of around ten: $\tau_{fast}$ = 5.29 ± 0.20 ms, and $\tau_{slow}$ = 40.30 ± 2.61 ms (*Figure 4D*). The relative fraction of the fast component was 77.21 ± 4.33%, indicating that the fast component of evoked release is dominant in *C. elegans* cholinergic NMJs. To assess the specific contribution of UNC-13L localization in SV release kinetics, we analyzed *unc-13(s69)* null animals expressing each UNC-13L variant. The time constants in *unc-13(s69); Si(UNC-13L)* were faster than those in wild type (*Figure 4D*), suggesting that other endogenous UNC-13 isoforms contribute to SV release with a slower kinetics. The time constants and the relative fraction of the fast component in *unc-13(s69); Si(UNC-13L$^{C2A-}$)* were both significantly changed, compared to those in *unc-13(s69); Si(UNC-13L)* (*Figure 4D*). *unc-13(s69); Si(UNC-13L$^{N-}$)* animals showed a more prolonged SV release. The time constants and the decay time of eEPSC were further affected, compared to *unc-13(s69); Si(UNC-13L$^{C2A-}$)* (*Figure 4C,D*). In *C. elegans* NMJs, the decays of tonic excitatory postsynaptic current (tEPSC), which represent spontaneous release of individual SVs, are general short (*Figure 4C*, *Figure 5—figure supplement 1A*). Moreover, the decay times of tEPSC are not altered in UNC-13L transgenic lines that have prolonged eEPSCs (see below, and [*Hu et al., 2013*]). Therefore, the observed changes in the evoked release kinetics are unlikely due to the kinetic change of postsynaptic ACh receptor response; and instead, the slower decay time of eEPSC reflects desynchronisation of presynaptic release. These results show that the $C_2A$ domain of UNC-13L is required for the fast release kinetics of SVs, and additional N-terminal protein sequences of UNC-13L further contribute to accelerating the $Ca^{2+}$-triggered evoked release.

Interestingly, *Si(UNC-13L$^{N-}$)* transgene rescued the amplitude of eEPSCs in *unc-13(s69)* to the wild type level, while *Si(UNC-13L$^{C2A-}$)* showed a partial rescue (*Figure 4B*). To address what might account for this difference, we performed recordings using sucrose application. We found that *Si(UNC-13L$^{N-}$)* showed significantly increased sucrose evoked SV release, comparing to *Si(UNC-13L)* and *Si(UNC-13L$^{C2A-}$)* (*Figure 4E*, *Figure 4—figure supplement 2*). Since UNC-13L$^{N-}$ shows diffused localization throughout presynaptic axons, the rescue of eEPSC amplitude by *Si(UNC-13L$^{N-}$)* likely reflects recruitment of SVs located distally from active zones. Together, with the enhanced slow release in *unc-13(s69);*

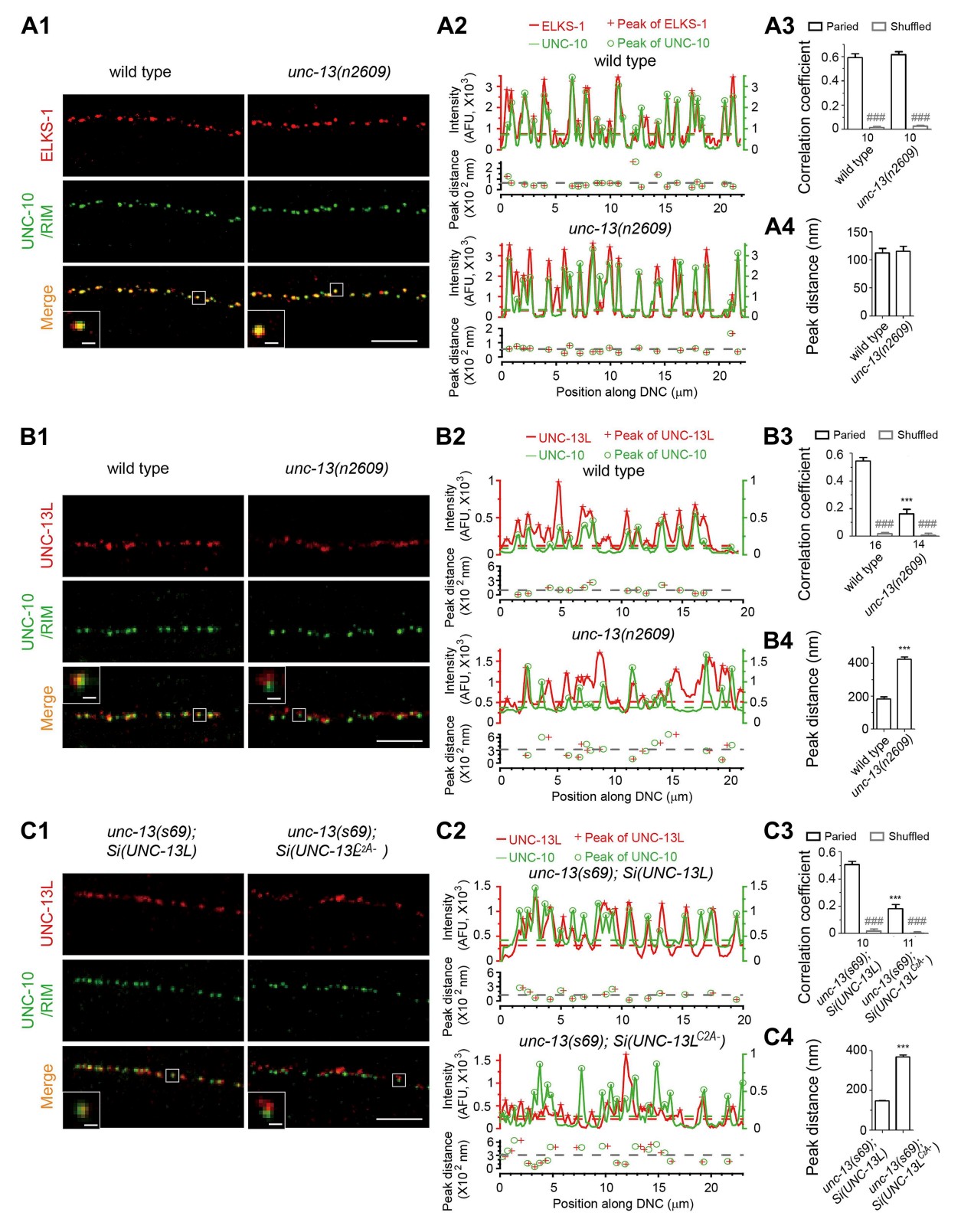

**Figure 3**. The $C_2A$ domain of UNC-13L is required for the precise localization of UNC-13L at active zones. (**A1**) Representative confocal Z-stack images of co-immunostaining for ELKS-1 and UNC-10/RIM from wild type and *unc-13(n2609)*. (**A2**) Average fluorescence intensities in six-pixel wide regions along a line drawn down the dorsal nerve cord (DNC) shown in **A1** corresponding to ELKS-1 and UNC-10/RIM signals. Peaks from ELKS-1 and UNC-10/RIM

*Figure 3. Continued on next page*

*Figure 3. Continued*

signals are differentially marked. The distances between the nearest peaks from fluorescence traces of ELKS-1 and UNC-10/RIM, which are less than 800 nm, are plotted against the positions of peaks along the line drawn down the DNC. The color broken lines indicate intensity thresholds to detect peaks from corresponding channels. The grey broken line indicates the average peak distance from that sample. (**A3**) Average pixel-by-pixel fluorescence intensity correlation coefficients between paired signals or shuffled data from ELKS-1 and UNC-10/RIM in wild type and *unc-13(n2609)*. (**A4**) Summary of the peak distances between ELKS-1 and UNC-10/RIM signals in wild type and *unc-13(n2609)*. (**B1-4**) Representative confocal Z-stack images (**B1**), average pixel-by-pixel fluorescence intensity correlation coefficients (**B3**), peak distance calculation from images shown in **B1** (**B2**) and summary (**B4**) of co-immunostaining for UNC-13L and UNC-10/RIM from wild type and *unc-13(n2609)*. (**C1-4**) Representative confocal Z-stack images (**C1**), average pixel-by-pixel fluorescence intensity correlation coefficients (**C3**), peak distance calculation from images shown in **C1** (**C2**) and summary (**C4**) of co-immunostaining for UNC-13L and UNC-10/RIM from *unc-13(s69); Si(UNC-13L)* and *unc-13(s69); Si(UNC-13L$^{C2A-}$)*. Scale bar: 5 μm in pictures and 0.5 μm in inserts for **A1**, **B1** and **C1**. For each intensity correlation comparison, a shuffled data set was also used to calculate the extent of random correlation between images (see 'Materials and methods'). AFU, arbitrary fluorescence units. The number of animals analyzed is indicated for each genotype. Error bars indicate SEM. Statistics, two-tailed Student's *t* test. \*\*\*p<0.001 for comparison between genotypes; ###p<0.001 for comparison between paired data set and shuffled data set for each genotype.

The following figure supplements are available for figure 3:

**Figure supplement 1**. Loss of C$_2$A domain does not change the co-localization between Ca$^{2+}$ channel and UNC-10/RIM.

**Figure supplement 2**. Presynaptic localization of UNC-13 is not solely dependent on UNC-10/RIM.

---

*Si(UNC-13L$^{N-}$)*, these observations indicate that these SVs are competent for release, but with lower release probability and slower release kinetics. *unc-13(s69); Si(UNC-13L$^{N-}$)* animals showed significantly slower locomotion than *unc-13(s69)* expressing either full-length UNC-13L or UNC-13L$^{C2A-}$ (*Figure 4—figure supplement 1*), suggesting that SV release probability and kinetics, rather than the total vesicle supply in RRP, are functional relevant determinants for synaptic transmission efficiency in these cholinergic neuromuscular junctions.

Lastly, as a further test to our conclusion that the distance of UNC-13L to calcium entry site directly influences SV release property, we performed recordings of *unc-13(n2609)* in 5 mM extracellular Ca$^{2+}$ concentration. While high [Ca$^{2+}$]$_{ex}$ resulted in an increase in eEPSC amplitudes in both wild type and *unc-13(n2609)* mutants, 5 mM [Ca$^{2+}$]$_{ex}$ had a stronger effect in *unc-13(n2609)* than in wild type animals (*Figure 4—figure supplement 3A*). Furthermore, we analyzed cumulative charge transfer kinetics of *unc-13(n2609)* (*Figure 4—figure supplement 3B*). In normal 2 mM [Ca$^{2+}$]$_{ex}$, *unc-13(n2609)* showed release kinetics defects similar to *unc-13(s69); Si(UNC-13L$^{C2A-}$)*. 5 mM [Ca$^{2+}$]$_{ex}$ increased the fraction of fast component in *unc-13(n2609)*, compared to wild type animals, although the time constant of fast component in *unc-13(n2609)* remains slower than that in wild type. Nonetheless, this result shows that higher [Ca$^{2+}$]$_{ex}$ can compensate for the lengthened distance to UNC-13L to calcium microdomain.

## The C$_2$A domain of UNC-13L has a specific role in spontaneous release

UNC-13 is also essential for spontaneous release (*Richmond et al., 1999*). To analyze the effects of UNC-13L active zone localization on spontaneous release, we recorded tonic excitatory post-synaptic currents (tEPSCs) from the cholinergic motor neurons. *unc-13(n2609)* mutants showed a strong reduction in tEPSC frequency, compared to wild type (*Figure 5A*). The amplitude and kinetics of tEPSCs was not altered (*Figure 5—figure supplement 1A*). Similarly, reduced tEPSC frequency was also observed in *unc-13(s69); Si(UNC-13L$^{C2A-}$)*, comparing to *unc-13(s69); Si(UNC-13L)*. Overexpression of UNC-13L$^{C2A-}$ in *unc-13(s69)* did not fully rescue the defects of tEPSC frequency, while overexpression of UNC-13L displayed an enhanced tonic release (*Figure 5—figure supplement 1C*). We also recorded tEPSCs in *unc-13(s69); Si(UNC-13L$^{N-}$)* animals, in which UNC-13 proteins are diffuse throughout the axon, and observed reduced tEPSC frequency to a level similar to that in *unc-13(s69); Si(UNC-13L$^{C2A-}$)* (*Figure 5A*), indicating that the C$_2$A domain alone accounts for the specific effect of the active zone localized UNC-13L in tonic release. Since loss of the C$_2$A domain caused UNC-13L to be shifted away from the center of the active zone (*Figures 3*) and SVs docked in regions distal to the active zone were competent for release (*Figures 1F and 4E*), these results suggest that a major proportion of tonic release may occur in regions proximal to the active zone. To test this idea further, we examined double mutants of *unc-13(n2609)* and *cpx-1/complexin*. CPX-1/complexin is a key regulator of SV release by acting as a clamp on SNARE (*Reim et al., 2001*; *Xue et al., 2007*; *Giraudo et al., 2009*; *Maximov et al., 2009*). Loss of CPX-1 significantly enhances tEPSC frequency (*Hobson et al., 2011*;

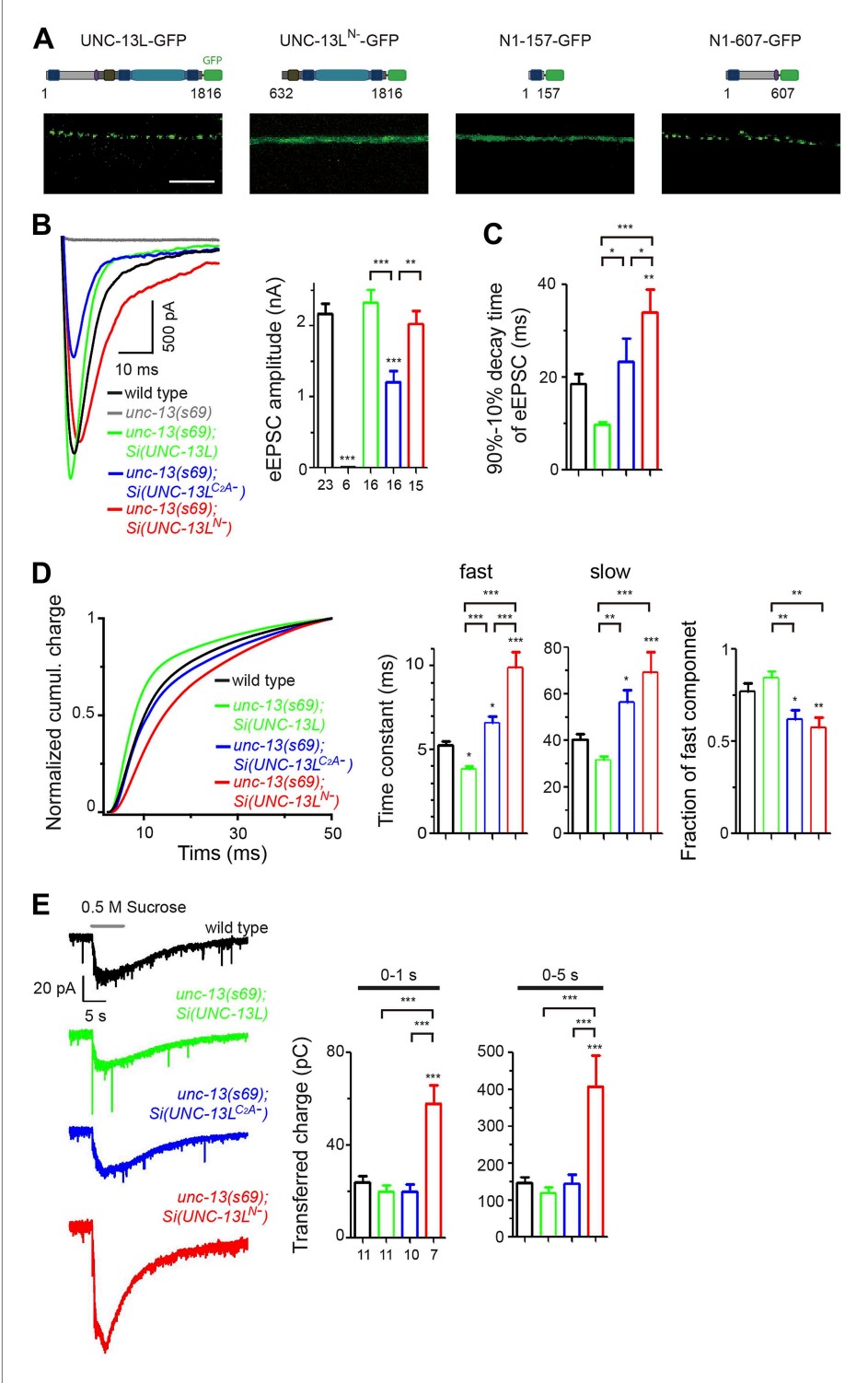

**Figure 4**. The N-terminal region of UNC-13L determines the presynaptic active zone localization of UNC-13L and is necessary for fast kinetics of evoked release. (**A**) Schematics and images in dorsal nerve cords of GFP tagged full length UNC-13L, UNC-13L$^{N-}$ lacking the entire N-terminal region (amino acids 632–1816), N-terminal amino acids 1–157 fragment and N-terminal amino acids 1–607 fragment driven by pan-neuronal promoter P*rgef-1*. Scale bar: 5 µm. (**B** and **C**) Average recording traces, mean peak amplitudes (**B**) and 90–10% decay time (**C**) of eEPSCs in animals of genotype indicated. The wild type data are the same data set in *Figure 1*. (**D**) The normalized cumulative charges of eEPSCs within 50 ms after electrical stimuli, time constants fitted with a double exponential

*Figure 4. Continued on next page*

*Figure 4. Continued*

function and relative fractions of fast component in animals of genotypes indicated. (**E**) Average recording traces (left), and transferred charges (right) of 0.5 M hypertonic sucrose solution induced vesicle release in animals of genotype indicated. The wild type data are the same data set in *Figure 1*. The number of animals analyzed is indicated for each genotype. Error bars in **B**–**E** indicate SEM. Statistics, one way ANOVA. ***$p < 0.001$; **$p < 0.01$; *$p < 0.05$.

The following figure supplements are available for figure 4:

**Figure supplement 1**. Locomotion speeds of *unc-13(s69)* rescue strains.

**Figure supplement 2**. Ratios of mean charge transfers during eEPSC and during sucrose application.

**Figure supplement 3**. Higher $[Ca^{2+}]_{ex}$ partially rescue eEPSC of *unc-13(n2609)*.

*Martin et al., 2011*), but it is not clear where within the synapse the increased tEPSC events occur. We found that tEPSC frequency in *cpx-1(ok1552) unc-13(n2609)* double mutants was significantly reduced, compared to *cpx-1(ok1552)* mutants (*Figure 5B*), indicating that the enhanced spontaneous release caused by loss of complexin requires the function of the UNC-13 $C_2A$ domain. We next recorded evoked release in *cpx-1(ok1552) unc-13(n2609)* double mutants. *cpx-1(ok1552)* single mutant showed dramatically reduced eEPSCs (*Figure 5C*), in part due to loss of a facilitating function of complexin on SV release (*Reim et al., 2001*; *Xue et al., 2007*; *Maximov et al., 2009*; *Hobson et al., 2011*; *Martin et al., 2011*). The amplitude of eEPSC in *cpx-1(ok1552) unc-13(n2609)* double mutants was significantly reduced, compared to wild type or *unc-13(n2609)*, but was moderately increased, compared to *cpx-1* single mutants. Analysis of charge transfer further showed a noticeable increase primarily in the fast phase of release, within 20 ms after electrical stimulation (*Figure 5D*). Based on these observations, we infer that in these cholinergic synapses SV populations involved in spontaneous release may be mainly from the region proximal to the active zone, which, in *cpx-1(ok1552) unc-13(n2609)* mutants, were converted to account for the fast phase of evoked release.

## Acute inactivation of UNC-13L or UNC-13L$^{N-}$ preferentially inhibits the fast or slow phase of evoked releases, respectively

To further address the temporal and spatial requirement of active zone localization of UNC-13L in SV exocytosis, we next employed the InSynC (Inhibition of Synapses with CALI, for Chromophore-assisted light inactivation) technique (*Lin et al., 2013*). This method takes advantage of the singlet oxygen production by the genetically encoded photosensitizer miniSOG (mini singlet oxygen generator) to acutely ablate tagged proteins in vivo upon blue light illumination. We constructed miniSOG tagged UNC-13L, which localizes to the active zone, and UNC-13L$^{N-}$, which is diffuse in axons (*Figure 4A*). In *unc-13(s69)*, both UNC-13L-miniSOG and UNC-13L$^{N-}$-miniSOG rescued the paralysis to different degrees (*Figure 6—figure supplement 1A*), indicating miniSOG tagged UNC-13L proteins are functionally incorporated into the SV release apparatus. Upon pulsed blue-light illumination, both miniSOG transgenic animals exhibited fast paralysis to a similar degree (*Figure 6—figure supplement 1A*), indicating miniSOG-mediated chromophore-assisted light inactivation (CALI) can inactivate UNC-13L and UNC-13L$^{N-}$ equally effectively. By NMJ recordings, without blue light, both UNC-13L-miniSOG and UNC-13L$^{N-}$-miniSOG fully rescued the amplitude of eEPSCs (*Figure 6—figure supplement 1B*). CALI by 2–3 min blue light illumination resulted in a severe inhibition of eEPSCs in both transgenic animals, while the same illumination had little effect on wild type animals expressing miniSOG tagged free YFP (miniSOG-Citrine) (*Figure 6A,B*).

We next tested whether acute photo-inactivation of UNC-13L-miniSOG and UNC-13L$^{N-}$-miniSOG could cause specific inhibition of synapse transmission in wild type animals. Our assumption is that transgenically over-expressed UNC-13L-miniSOG or UNC-13L$^{N-}$-miniSOG would interact with native protein interacting partners by competing with endogenous UNC-13. Wild type animals carrying UNC-13L-miniSOG transgenes showed rapid movement impairment upon CALI by blue light (*Figure 6—figure supplement 1A*). Notably, animals with UNC-13L-miniSOG showed much slower movement than those with UNC-13L$^{N-}$-miniSOG (*Figure 6—figure supplement 1A*), supporting our assumption that UNC-13L-miniSOG and UNC-13L$^{N-}$-miniSOG are incorporated into the endogenous SV release

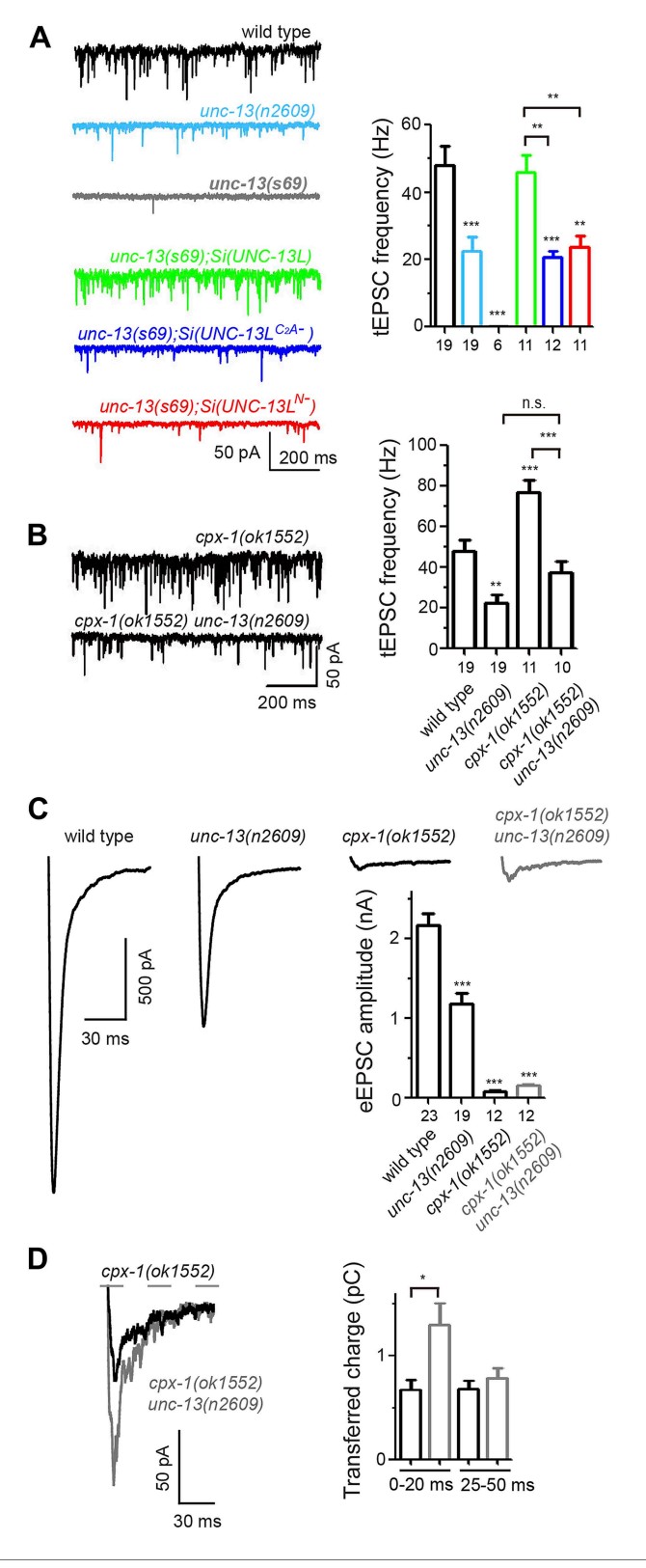

**Figure 5**. The C₂A domain of UNC-13L is required for tonic synaptic vesicle release. (**A** and **B**) Representative recording traces (left) and summary (right) of tEPSC frequency in animals of genotype indicated. (**C**) Average recording traces and mean peak amplitudes of eEPSCs in animals of genotype indicated. (**D**) Superposed average

*Figure 5. Continued on next page*

*Figure 5. Continued*

recording traces, 0–20 ms transferred charge and 25–50 ms transferred charge of eEPSCs from *cpx-1(ok1552)* and *cpx-1(ok1552) unc-13(n2609)*. The number of animals analyzed is indicated for each genotype. Error bars indicate SEM. Statistics, one way ANOVA for multiple groups in **A**–**C** and two-tailed Student's *t* test in D. ***$p<0.001$; **$p<0.01$; *$p<0.05$.

The following figure supplements are available for figure 5:

**Figure supplement 1**. Tonic EPSC amplitudes and decay times of *unc-13(s69)* rescue strains and *cpx-1* mutants, and the rescue effects of overexpression of UNC-13L and UNC-13L$^{C2A-}$ on tEPSC in *unc-13(s69)*.

apparatus in different subsynaptic domains. We then performed NMJ recordings. Without blue light treatment, overexpression of UNC-13L-miniSOG in wild type animals caused increased eEPSC amplitude and charge transfer in the fast phase of release, compared to control animals expressing miniSOG-Citrine (*Figure 6A,B*). CALI of UNC-13L-miniSOG dramatically reduced the amplitude of eEPSCs, and resulted in a strong decrease in the transferred charge of the fast phase, but little effect on the slow phase of eEPSCs (*Figure 6B*). In contrast, overexpression of UNC-13L$^{N-}$-miniSOG in wild type animals, without blue light illumination, resulted in a large slow phase of evoked release (*Figure 6A–B*), which is consistent with the report that UNC-13L$^{N-}$ is able to induce release competent SVs with slow release kinetics (*Hu et al., 2013*). CALI of UNC-13L$^{N-}$-miniSOG caused a mild reduction in the amplitude and the fast phase of eEPSCs, but nearly abolished the slow phase. Importantly, inactivation of UNC-13L-miniSOG resulted in a much slower 90–10% decay time of eEPSCs than the control animals expressing free miniSOG-Citrine, whereas inactivation of UNC-13L$^{N-}$-miniSOG had an opposite effect (*Figure 6C*). Since UNC-13L and UNC-13L$^{N-}$ reside in different subdomains of synapses and likely interact with the release machinery for different pools of SVs, we interpret that the differential effects of acute ablation of UNC-13 protein variants reflect the consequence of inhibiting or damaging themselves and their immediately associated protein interacting partners that are necessary for their action in situ. These results are consistent with the conclusion that the active zone localization of UNC-13L, hence close proximity to the $Ca^{2+}$ entry site, is critical for the fast phase of evoked release.

We further tested the specificity of acute ablation of UNC-13 functional complex in spontaneous release. All transgenic animals showed stable levels of tEPSC frequency after 2 min illumination (*Figure 6D,E*). In *unc-13(s69)* mutants, UNC-13L-miniSOG fully rescued tEPSC frequency, while UNC-13L$^{N-}$-miniSOG partially rescued it (*Figure 6—figure supplement 1C*). Blue light illumination caused a strong inhibition on both rescued lines. In wild type background, inactivation of UNC-13L-miniSOG dramatically reduced the frequency of tEPSCs by 70% in the presence of endogenous proteins, compared to the pre-light condition (*Figure 6D,F*). Inactivation of UNC-13L$^{N-}$-miniSOG had a weak effect on the frequency of tEPSCs. Together, these analyses suggest that UNC-13L$^{N-}$-miniSOG, being diffusely localized in axons, has a less role in interacting with the release apparatus for tonic release, and provide further support for the conclusion that the precise localization of UNC-13L to the active zone is crucial for spontaneous release.

## Inducible ablation of UNC-13L reversibly modulates the epileptic-like convulsive behavior of *acr-2(gf)*

We isolated *unc-13(n2609)* allele as a genetic suppressor of the behavior deficits caused by *acr-2(n2420gf)*, which causes over-excitation in the locomotion circuit and exhibits spontaneous and frequent whole body muscle contractions (*Jospin et al., 2009*) (*Figure 7A*). A similar amino acid change in a non α-subunit of acetylcholine receptors in the human brain has been reported to cause epilepsy (*Phillips et al., 2001*). *unc-13(n2609)* strongly suppresses *acr-2(gf)*-induced convulsions (*Figure 7A*). This behavioral suppression is likely due to reduced over-excitation as *unc-13(n2609); acr-2(gf)* double mutants showed reduced tEPSCs compared to *acr-2(gf)* (*Figure 7—figure supplement 1*). Interestingly, *unc-13(n2813),* which contains a missense mutation in the C-terminal MUN domain and reduces SV priming to less than half of that in wild type (*Richmond et al., 1999*), showed much weaker suppression on convulsions in *acr-2(gf)* animals (*Figure 7A*). These observations are consistent with our overall conclusion that the $C_2A$ domain-containing full length of UNC-13L and the C-terminal region of UNC-13L mediate different modes of synaptic transmission, and suggest that specific modes of synaptic transmission involving the $C_2A$ domain may underlie synaptic dysfunction in *acr-2(gf)* animals.

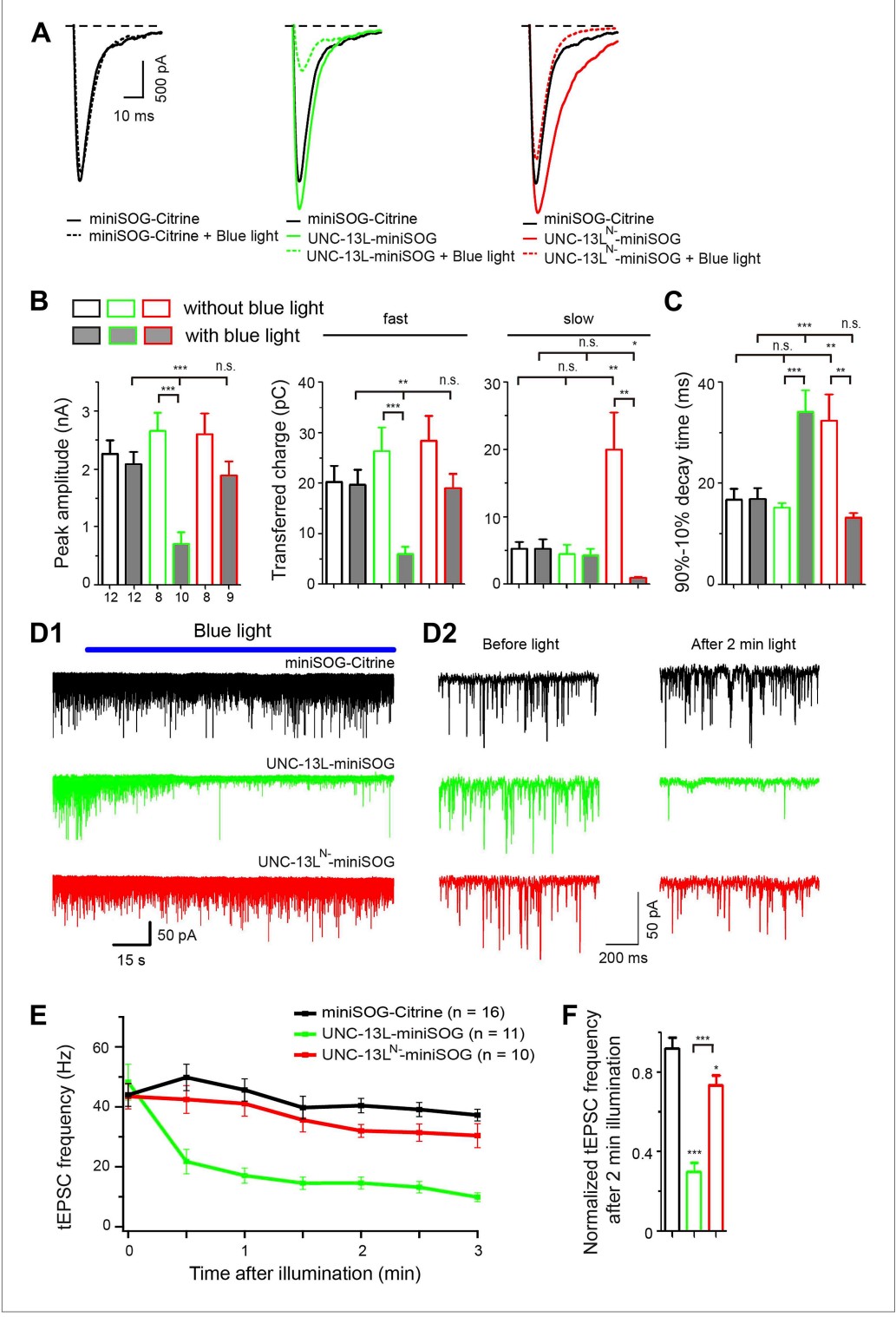

**Figure 6**. MiniSOG-mediated acute abalation supports a specific role of UNC-13L in fast phase of evoked release and in tonic release. (**A**) Average recording traces of eEPSCs in animals of genotype indicated without or with blue light treatment. (**B** and **C**) Summaries of the peak amplitude, transferred charge of fast component and slow component (**B**) and 90–10% decay time (**C**) of eEPSCs from genotypes shown in **A**. (**D1–2**) Representative recording traces of tEPSC with blue light illumination (**D1**), enlarged recording traces in 1 s duration before and after 2 min blue light illumination (**D2**) in animals of genotype indicated. (**E** and **F**) Average frequencies of tEPSCs during blue
*Figure 6. Continued on next page*

*Figure 6. Continued*

light illumination (**E**) and normalized tEPSC frequencies after 2 min illumination to mean tEPSC frequencies before illumination (**F**) from genotypes shown in **D**. The number of animals analyzed is indicated for each genotype. Error bars indicate SEM. Statistics, one way ANOVA among different genotypes and two-tailed Student's *t* test for a given genotype with or without blue light. ***p<0.001; **p<0.01; *p<0.05; n.s., not significant.
The following figure supplements are available for figure 6:

**Figure supplement 1**. Effects of acute miniSOG-mediated CALI of UNC-13L and UNC-13L$^{N-}$ on locomotion speeds and on SV release in *unc-13(s69)*.

---

To further investigate the contributions of the N- and C-terminal regions of UNC-13L to *acr-2(gf)*-induced convulsions, we expressed UNC-13L variants in *acr-2(gf)* animals. Interestingly, overexpression of UNC-13L or genomic *unc-13* in *acr-2(gf)* animals exacerbated convulsions (***Figure 7A***). In contrast, overexpression of UNC-13L$^{N-}$ in *acr-2(gf)* animals had no augmentative effect on convulsions. Lastly, to address whether the suppression of *acr-2(gf)*-induced convulsions by *unc-13* mutant is due to an acute effect to inhibit over-excitation, we introduced UNC-13L-miniSOG and UNC-13L$^{N-}$-miniSOG into *acr-2(gf)* animals. Upon blue light illumination, UNC-13L-miniSOG expressing animals showed a strong suppression of convulsions, while UNC-13L$^{N-}$-miniSOG expressing animals continued to convulse

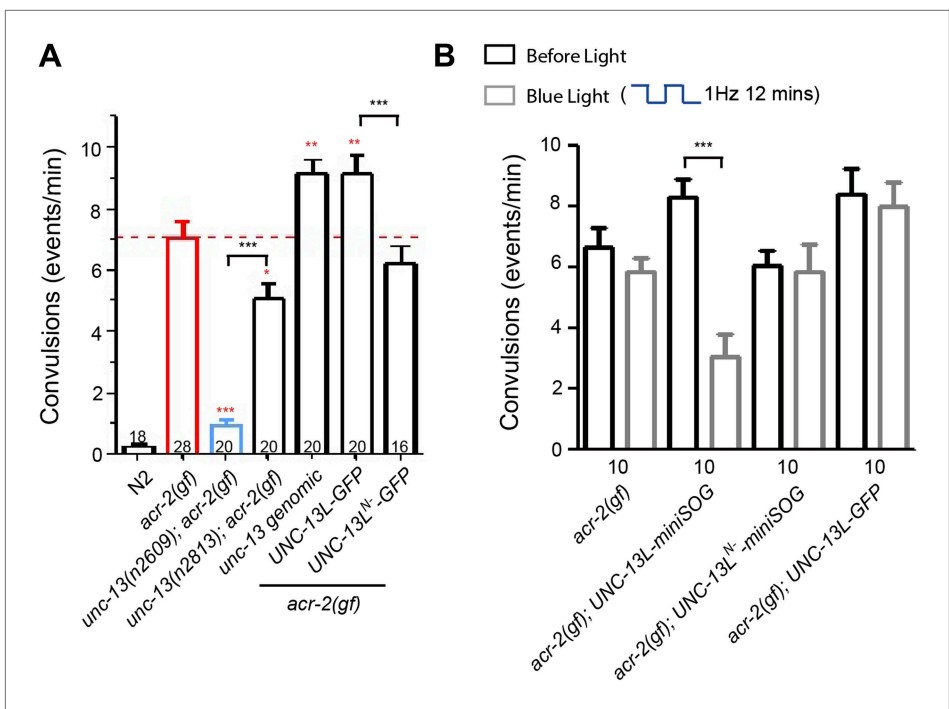

**Figure 7**. The C$_2$A domain-containing N-terminal region of UNC-13L is required for *acr-2(gf)*-induced epileptic-like convulsions. (**A**) Summary of the suppression of *unc-13(n2609)*, *unc-13(n2813)* on *acr-2(gf)*-induced convulsions, and the effects of *unc-13* genomic DNA cosmid C44E1, UNC-13L and UNC-13L$^{N-}$ transgene on convulsions in *acr-2(gf)* mutants. ***p<0.001, **p<0.01 and *p<0.05 (red), compared to *acr-2(gf)*. (**B**) Summary of the effects of blue light treatment on convulsions in L4 stage animals of genotype indicated. The number of animals analyzed is indicated for each genotype. Error bars indicate SEM. Statistics, one way ANOVA in **A** and paired two-tailed Student's *t* test for a given genotype with or without blue light in **B**. ***p<0.001; **p<0.01.
The following figure supplements are available for figure 7:

**Figure supplement 1**. Tonic release in *acr-2(gf)* mutants is reduced by *unc-13(n2609)*.

**Figure supplement 2**. Recovery of convulsions in *acr-2(gf)*; *UNC-13L-miniSOG* after blue light treatment.

similarly to *acr-2(gf)* animals (**Figure 7B**). After blue light treatment, *acr-2(gf)* animals expressing UNC-13L-miniSOG gradually restored convulsions, showing a full recovery after 16 hr (**Figure 7— figure supplement 2**), which may reflect the time course of presynaptic UNC-13L protein turnover. This latter analysis also shows that the effect of miniSOG-mediated CALI is reversible, and suggests the possibility of temporal interference of specific aspects of synaptic transmission in controlling synapse dysfunction underlying some neurological disorders.

## Discussion

Differential expression and function of UNC-13/Munc13 isoforms endow synapses with distinct release properties (**Augustin et al., 2001**; **Rosenmund et al., 2002**). Several recent studies have begun to unveil the function specificity mediated by the N-terminal domains in different Munc13 isoforms (**Deng et al., 2011**; **Chen et al., 2013**; **Hu et al., 2013**; **Lipstein et al., 2013**). The non-calcium binding $C_2A$ domain of UNC-13/Munc13 serves as protein interacting domain to bind itself or the active zone protein RIM (**Betz et al., 2001**; **Lu et al., 2006**). In this study, taking advantage of the *unc-13(n2609)* mutation as well as single-copy expression of UNC-13L variants in *unc-13* null mutants, we have uncovered specific roles of the $C_2A$ domain in SV release probability and spontaneous release. The precise active zone localization depends on the $C_2A$-containing N-terminal region unique to UNC-13L, and directly contributes to SV release kinetics. Our data support a conclusion that the proximity of UNC-13/Munc13 to the $Ca^{2+}$ entry site plays a critical role in SV release probability and release kinetics, and also suggest that spontaneous release and the fast phase of evoked release may share a common pool of synaptic vesicles at the active zone.

Previous studies, largely based on overexpression of mutant Munc13/UNC-13 proteins in cultured neurons or transgenic animals, have suggested that N-terminal $C_2A$ domain is necessary for their localization at the active zone (**Andrews-Zwilling et al., 2006**; **Hu et al., 2013**). Here, we show that lack of $C_2A$ domain causes a delocalization of UNC-13L from UNC-10/RIM, resulting in a shift of UNC-13L from the center of the active zone. Homodimerization of the $C_2A$ domain of Munc13 is recently shown to inhibit its function in SV priming, whereas RIM binding to the $C_2A$ domain converts this priming-inhibitory state to a priming-promoting state (**Deng et al., 2011**). A monomeric Munc13 lacking the $C_2A$ domain-containing N-terminal region can rescue the priming defects of synapses lacking majority of Munc13, but does not fully rescue evoked release (**Deng et al., 2011**). Consistent with this study, we find that SV priming is normal in synapses lacking specifically the $C_2A$ domain of UNC-13L. We further show that lack of $C_2A$ domain specifically reduces the release probability of SV release. Based on the immunostaining of UNC-13L and ultrastructural analysis of *unc-13(n2609)*, we think that this effect is likely due to both mispositioning of the remaining UNC-13L in the active zone as well as a mild effect on SV docking at the proximal region to active zones.

The SV release kinetics has been primarily attributed to the intrinsic $Ca^{2+}$ sensitivity modulated by distinct $Ca^{2+}$ sensor proteins (**Südhof, 2012a**). The distance between SV release sites to $Ca^{2+}$ influx sites also significantly influences release kinetics (**Neher and Sakaba, 2008**). Among the core SV fusion apparatus, including SNARE and Munc18 proteins, UNC-13/Munc13 exhibits the most restricted localization at the active zone (**Südhof, 2012b**). In *C. elegans* the precise active zone localization of UNC-13L is regulated by the $C_2A$ domain-containing N-terminal region (this study, and [**Hu et al., 2013**]). Overexpression of a chimeric protein with only the $C_2A$ domain attached to the C-terminal common region of UNC-13 shortens the latency of SV release (**Hu et al., 2013**). We find that lacking the $C_2A$ domain of UNC-13L causes reduced amplitude of evoked release, which can be partially suppressed by increasing extracellular calcium, supporting that the primary defect in $unc-13^{C2A-}$ animals is the increased distance between UNC-13 and calcium entry site. Our results that complete loss of the N-terminus of UNC-13L dramatically alters the time constants of charge transfer are completely consistent with the recent report that the non-active zone localized UNC-13S or the N-terminus truncated UNC-13L mediate slow evoked release of SVs (**Hu et al., 2013**). Together, our two studies demonstrate that the localization of UNC-13 at the active zone is a crucial determinant for $Ca^{2+}$ influx accelerating SV release. The functional effect of differential localization of murine Munc13s has also been tested in the calyx of Held where Munc13-2/3 isoforms, which are much less localized in the active zone, are selectively involved in slowly releasing vesicles pool, while $C_2A$ domain-containing Munc13–1 is the dominant priming factor by electrophysiological recording (**Chen et al., 2013**).

Investigation of synaptic transmission in *C. elegans* has traditionally relied on the use of genetic mutations that perturb gene function at birth. Comparing to the previously reported synapse CALI methods

(*Marek and Davis, 2002*; *Snellman et al., 2011*), the miniSOG mediated InSynC technology has the advantage to reversibly remove protein function in vivo without addition of exogenous cofactors (*Lin et al., 2013*). The molecular basis of InSynC in wild type animals remains to be investigated, and likely involves dominant negative effects on the SV release apparatus containing miniSOG-tagged proteins. Our study here provides further evidence for the specificity and utility of this methodology. UNC-13L-miniSOG and UNC-13L$^{N-}$-miniSOG are differentially localized in presynaptic terminals. UNC-13L-miniSOG is likely to be associated with proximal SV release apparatus close to $Ca^{2+}$ entry sites, while UNC-13L$^{N-}$-miniSOG can interact with distal SV release apparatus. We find that inactivation of UNC-13L-miniSOG and UNC-13L$^{N-}$-miniSOG preferentially inhibited the fast phase and the slow phase of evoked release in wild type background, respectively. Moreover, the fact that animals can recover after CALI indicates significant level of protein turnover at synapses, suggesting possible applications of miniSOG-based optogenetic tools in investigating protein homeostasis in vivo and in situ.

Our analysis that the $C_2A$ domain of UNC-13L has a specific role in spontaneous release also sheds some light to the source of SV pools in distinct release mode. Since the discovery of spontaneous release by Fatt and Katz (*Fatt and Katz, 1952*), many studies have revealed that spontaneous release contributes to physiological processes. For example, spontaneous release regulates the initiation of action potential in hippocampal pyramidal neurons and firing rates in cerebellar interneurons (*Carter and Regehr, 2002*; *Sharma and Vijayaraghavan, 2003*), influences dendritic spine morphology (*McKinney et al., 1999*), inhibits local dendritic protein translation (*Sutton et al., 2006*) and modulates homeostatic synaptic plasticity (*Aoto et al., 2008*). Several molecules such as Doc2b (*Groffen et al., 2010*) and Vti1a (*Ramirez et al., 2012*) appear to function preferentially in spontaneous release. However, the question of whether spontaneous release and evoked release utilize distinct SV populations remains difficult to resolve (*Alabi and Tsien, 2012*). Studies using the similar preparations and measurements often reach different conclusions (*Sara et al., 2005*; *Groemer and Klingauf, 2007*). We find that removing $C_2A$ domain, as in *unc-13(n2609)* animals, specifically reduces spontaneous release and the fast phase of evoked release to about 50% of wild type animals, and suppresses the increased spontaneous release in *cpx-1(null)* mutants. The reduced tonic release in *unc-13(n2609) cpx-1(null)* is accompanied with a noticeably enhanced fast phase of evoked release. CALI of UNC-13L shows a strong effect of the active zone localized UNC-13L in spontaneous release and the fast phase of evoked release, but not slow phase. In contrast, UNC-13L$^{N-}$ shows diffuse distribution and accounts for an enhanced slow phase of evoked release, and rescues the spontaneous release defect of *unc-13(null)* to a similar level to that of UNC-13L$^{C2A-}$. These results suggest that SVs associated with diffused UNC-13L$^{N-}$, the majority of which may be positioned distally from the active zone, mainly undergo evoked release with slow kinetics, and may not contribute to spontaneous release. Our data are generally consistent with the idea that spontaneous release and evoked release use the same pool of SVs, and further suggest that at the *C. elegans* cholinergic NMJs SV populations involved in spontaneous release and the fast phase of evoked release may likely reside at regions proximal to the active zone (*Figure 8*).

## Materials and methods

### Genetics

*C. elegans* strains were maintained on Nematode Growth Medium (NGM) plates at room temperature (20–22°C) as described (*Brenner, 1974*). Double mutants were constructed following standard procedures, and genotypes were confirmed by allele-specific sequence polymorphism. *Supplementary file 1A and 1B* show the details for each mutation and strains; and *Supplementary file 1C* lists the genotypes of all transgenic strains.

*n2609* was isolated as a suppressor of the convulsion behavior caused by *acr-2(gf),* in a previously described EMS mutagenesis screen (*Jospin et al., 2009*). Genetic mapping placed *n2609* on chromosome I. Whole genome sequencing analysis (*Sarin et al., 2008*) revealed that *n2609* contains a single nucleotide C to T change in exon 3 of the *unc-13* long isoform transcript, changing UNC-13L glutamine 46 to a stop codon.

### Molecular biology and transgenes

Molecular biology was performed according to standard methods (*Sambrook et al., 1989*). The constructs of miniSOG tagged UNC-13 were made using the Gibson assembly method (*Gibson et al.,*

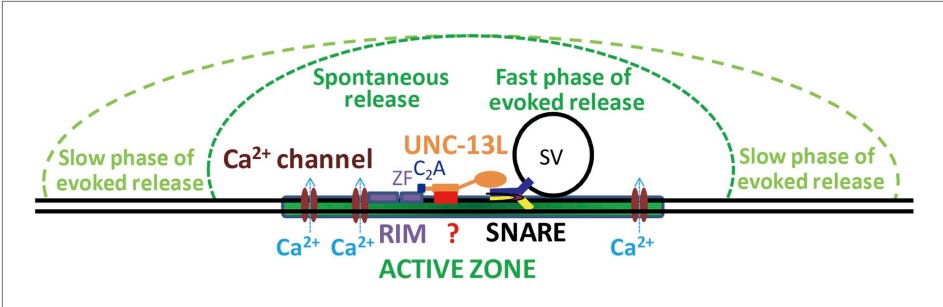

**Figure 8**. Model for the C$_2$A domain-containing N-terminal region of UNC-13L support spontaneous release and fast kinetics of evoked release. N-terminal sequences subsequent to the C$_2$A domain interact with unknown targets (represented by a question mark) to facilitate the presynaptic localization of UNC-13L. The C$_2$A domain binding to the zinc finger domain (ZF) of UNC-10/RIM promotes UNC-13L to be concentrated at active zones, where Ca$^{2+}$ channels reside. The UNC-13L anchored at active zones supports both spontaneous and the fast phase of evoked release. SVs in regions distal to the active zone are mainly involved in the slow phase of evoked release. The N-terminal region of UNC-13L facilitates the spontaneous and fast synchronous release by promotes UNC-13L and possibly SVs close to Ca$^{2+}$ influx sites.

*2009*). All other DNA expression constructs were made using Gateway cloning technology (Invitrogen, CA), following the manufacturer's procedures. DNA sequences were verified using restriction enzyme digestion and sequencing. *Supplementary file 1C* lists constructs and transgenes. Cosmid C44E1 was made by Sanger center, and obtained from James Rand (Oklahoma Medical Research Foundation). High-copy number transgenes were generated following standard procedures (*Mello et al., 1991*). In general, plasmid DNAs of interest were used at 10 ng/µl and co-injection markers P*ttx-3*-RFP at 50–90 ng/µl. For each construct, multiple independent transgenic lines were analyzed. Mos1-mediated single copy insertions were at the insert into *ttTi5605* site of chromosome II as described (*Frøkjær-Jensen et al., 2008*).

## Quantification of convulsion

L4 larvae were placed on freshly seeded NGM plates. The following day, individual young adults were transferred to fresh plates and recorded by video for 90 s. 8–10 animals were recorded for each genotype per trial and at least two trials were performed per genotype. A convulsion was defined as a visible sudden shortening in the animal's body length (*Jospin et al., 2009*).

## Quantification of locomotion

Locomotion speeds were measured using Worm Tracker 2.0 (W. Schafer's laboratory, MRC Laboratory of Molecular Biology, Cambridge, UK) (*Ben Arous et al., 2010*), and animals were prepared as described (*Qi et al., 2012*). If locomotion speeds were measured with food, the NGM plates were seeded with OP50 bacteria on the day before experiments and were kept at room temperature overnight. Immediately before transferring the worms, about 300 µl of 100 mM CuCl$_2$ was poured and swirled on the rim of the NGM plate to form a 'copper ring', and excessive CuCl$_2$ solution was removed. Individual young adults grown on an OP50 lawn were gently transferred to M9 solution using an eyelash. Any bacteria were rinsed off using an aspiration micropipette, and the worm was then transferred onto a fresh tracking plate using the same micropipette. The plate was placed on the tracker platform, and tracking started about 90 s after the puddle of M9 with the worm was absorbed into the agar and the worm had started crawling. Each tracking movie lasted 5 min with 10 frames per second. Movies were analyzed using the algorithms modified by Suk-Ryool Kang (Department of ECE, University of California, San Diego).

## Blue light treatment on live animals

Before transferring Larva 4 stage worms to plates, 3-cm NGM plates were spread with 15 µl concentrated OP50 to form thin OP50 lawn by waiting for a short while. About 100 µl of 100 mM CuCl$_2$ was poured and swirled on the rim of the plate to form a 'copper ring' to keep worms away from the edge of plates, and excessive CuCl$_2$ solution was removed. Plates containing worms were illuminated

with blue LED light source (460 nm, spectrum half width 27 nm, Prizmatix, Givat Shmuel, Israel). The diameter of illumination area is 5.8 cm to cover the entire surface of plates. The light intensity was measured to be 2.07 mW/mm² with a power sensor D10MM connected to an Optical Power Meter PM50 (Thorlabs, Newton, New Jersey, US). Animals were exposed to 1 Hz pulsed blue light for 12 min to avoid high heat accumulation on plates (*Qi et al., 2012*). The frequency of pulsed light is controlled by TTL signals provided with PASCO digital function generator PI-9587C (Roseville, California, US). Before the locomotion or convulsion measurement, worms were transferred back to normal OP50 seeded NGM plates for recovery for around 5–10 min after blue light treatment.

## Electron microscopy of synaptic vesicle distribution

Young adult worms were immobilized by high-pressure freezing at −176°C in the BAL-TEC HPM 010. Then frozen worms were freeze substituted in the Leica EM AFS2 system with 2% osmium tetroxide and 0.1% uranyl acetate in acetone for 4 days at −90°C and 16 hr at −20°C. After infiltration and embedding in Durcupan ACM resin blocks were polymerized at 60°C for 48 hr. Serial sections of 33 nm thicknesses were collected using the ultramicrotome Leica ULTRACUT UCT and stained for 5 min in 2.5% uranyl acetate in 70% methanol, followed after washing by 3 min in Reynold's lead citrate. All images of synapses with density from ventral nerve cord were obtained on a JEOL-1200 EX transmission electron microscope using Gatan 4 MP digital camera and DigitalMicrograph acquisition software. Distances from the edge of the dense projection to all docked vesicles along membrane were measured using ImageJ software. The distance from the dense projection to each docked vesicle was sorted into 33 nm bins. The number of vesicles in each bin was divided by the number of profiles to yield an average number of vesicles per profile in each bin to generate the histogram of docked vesicles. Only vesicles in profiles containing a dense projection were included. The histogram of docked synaptic vesicles was integrated and normalized to generate the accumulative fraction. Each contiguous set of serial profiles containing a dense projection was considered as a single data point, that is a synapse. The number of docked vesicles in specific regions (<165 nm, <231 nm, 232–330 nm and >330 nm) within each such set was divided by the number of profiles in the set, resulting in a number of docked vesicles per profile for that set. The mean and SEM of all data points within each genotype was determined and used to calculate p values in two-tailed Student's *t* test.

## Immunostaining and imaging

Whole-mount staining was conducted using 1% paraformaldehyde fixation as previously described (*Finney and Ruvkun, 1990*). Primary antibodies used were mouse anti-UNC-10/RIM (RIM2-s from Developmental Studies Hybridoma Bank, Iowa City, IA) (*Hadwiger et al., 2010*) at 1:3 dilution, rabbit anti-UNC-13 Rab598 at 1:35 ratio (gift from James Rand) (*Kohn et al., 2000*), rabbit anti-ELKS-1 Rb237 at 1:200 (gift from Michael Nonet) (*Deken et al., 2005*) and rabbit anti-GFP (A11122 from Invitrogen, CA) at 1:500. Secondary antibodies were goat anti-mouse Alexa Fluor 488 (A11001), goat anti-rabbit Alexa Fluor 594 (A11012), goat anti-mouse Alexa Fluor 594 (A11005) and goat anti-rabbit Alexa Fluor 488 (A11008) from Invitrogen, and used at 1:2000 dilution. Confocal images were taken on a Zeiss LSM 510 with 488 nm and 594 nm lasers. Laser output was set to 40% and transmission was optimized for detection and minimum bleed-through. Single 0.5 µm confocal planes were captured, merged using LSM software and exported as lsm file. To compare the correlation of signals from two channels, signals in each channel were separated with MetaMorph (Sunnyvale, CA) and exported as TIFF file. After thresholds were set, pixel-by-pixel intensity correlation analysis was performed and plotted automatically by Metamorph between two channels in the same animal (paired correlation). Since some fluorescence overlapping could have arisen by chance, the images from green and red channels were shuffled to determine green-red correlations between animals (shuffled correlation). In all cases, the shuffled correlation coefficient was nearly zero, confirming that the measured paired correlation is not due to chance. To calculate the distance from the center of a punctum in one channel to the center of the nearest punctum in another channel, average fluorescence intensities in six or eight-pixel wide segment regions along a line drawn down the dorsal nerve cord (DNC) were calculated by MetaMorph and plotted against the pixel position along the scan-line in IGOR Pro (WaveMetrics, Lake Oswego, OR). The peaks of each channel signal above the threshold were automatically found in IGOR, which representing the center of puncta. To determine the threshold, the lowest non-zero point of a given signal trace in the region without punctum fluorescence was identified in IGOR. The standard deviation (SD) of the signal trace within 600 nm region with the lowest non-zero point as the center was calculated.

The threshold was initially set as the value of 3.5-fold of SD plus the lowest non-zero value. If needed, the threshold was then manually adjusted to include all peaks from fluorescence puncta for the analysis. For each peak of RIM signal, the nearest peak from another channel with 800 nm was identified and the distance between the two peaks along scan-line was calculated.

## Electrophysiology

Neuromuscular dissection methods were adapted from previous studies (*Richmond et al., 1999*). Adult worms were immobilized on Sylgard-coated cover slips with cyanoacrylate glue. A dorsolateral incision was made with a sharp glass pipette and the cuticle flap was folded back and glued down to expose the ventral medial body wall muscles. The preparation was then treated by collagenase type IV (Sigma-Aldrich, St. Louis, MO) for ~ 30 s at a concentration of 0.4 mg/ml at room temperature.

The bath solution containing (in mM): 127 NaCl, 5 KCl, 26 $NaHCO_3$, 1.25 $NaH_2PO_4$, 2 $CaCl_2$, 4 $MgCl_2$, 10 glucose, and sucrose to 340 mOsm, bubbled with 5% $CO_2$, 95% $O_2$ at 20°C. The pipette solution containing (in mM): 120 $CH_3O_3SCs$, 4 CsCl, 15 CsF, 4 $MgCl_2$, 5 EGTA, 0.25 $CaCl_2$, 10 HEPES and 4 $Na_2ATP$, adjusted to pH 7.2 with CsOH. The extracellular calcium concentration is otherwise indicated if it is not 2 mM. Conventional whole-cell recordings from muscle cells were performed at 20°C with 2–3 MΩ pipettes. An EPC-10 patch-clamp amplifier was used together with the Patchmaster software package (HEKA Electronics, Lambrecht, Germany). Tonic EPSCs were recorded at −60 mV. For record evoked EPSCs, a second glass pipet filled with bath solutions was put on the ventral nerve cord as stimulating electrode. The stimulating electrode gently touched the anterior region of ventral nerve cord to form loose patch configuration, which is around 1 muscle distance from recording pipets. A 0.5 ms, 85 µA square current pulse was generated by the isolated stimulator (WPI A320RC, Sarasota, FL) as stimulus to obtain the maximal responses. For RRP depletion, the 0.5 M sucrose in bath solution was applied to ventral nerve cord near the recorded muscles by Picospritzer with 8 psi for 7 s. Under this prolonged stimulation protocol, we could observe the current decay over the stimulation window. The sucrose-evoked responses have been compensated for the basal activities by subtracting basal line current level prior to sucrose application. For miniSOG mediated CALI, illumination (15 or 30 mW/mm²) was provided with a Sutter Instrument Lambda LS fitted with a Lamda 10–2 filter wheel for shuttering (Novato, CA). The excitation light was filtered with an eGFP filter set with 480 nm excitation (Chroma N41012; Chroma, Bellows Falls, VT) and focused on the specimen with a 63 × water immersion objective (Olympus, Center Valley, PA). Light intensity was measured with a calibrated photometer with an integrating sphere detector (International Light Technologies, Newburyport, MA). A glass slide with a semi-spherical lens containing a water drop was placed under the objective and was used to direct the light into the integrating sphere. The area of illumination was measured with a stage micro-meter for the calculation of illumination intensity. The 2–3 min continuous blue light was illuminated on the prep including the ventral nerve cord and recorded muscle for miniSOG-mediated CALI. All current traces were imported to IGOR Pro (WaveMetrics, Lake Oswego, OR) for further analysis. The cumulative transferred charge of eEPSC was integrated over 50 ms after the electrical stimulation. The charge trace was fitted with a following double-exponential function to derive the time constant (τ) and size (*A*) of each component:

$$Q(t) = A_{fast} \cdot (1 - \exp(-(t - t_0)/\tau_{fast})) + A_{slow} \cdot (1 - \exp(-(t - t_0)/\tau_{slow}))$$

where $t_0$ is the time of electrical stimulation. The size of slow component ($A_{slow}$) was not directly used. Instead, we subtracted the $A_{fast}$ from the total amount of transferred charge within 50 ms.

## Statistical analysis

We used Graphpad Prism 5 (GraphPad Software, La Jolla, CA) to test significance. For comparisons of two groups, we used a two-tailed Student's *t* test. For comparisons involving multiple groups, we used one-way ANOVA and Newman-Keuls post hoc test. ***$p < 0.001$; **$p < 0.01$; *$p < 0.05$.

## Acknowledgements

We thank Yingchuan Qi for constructing some strains in early study of this work, Justine Levan for assistance in video recordings, Suk-Ryool Kang for modifying worm tracker software, James Rand for the cosmid C44E1 and UNC-13L antibodies, Michael Nonet for ELKS-1 antibodies, Terry Snutch for *vals33* marker. Some strains were provided by the *Caenorhabditis* Genetics Center, funded by the NIH

National Center for Research Resources. We thank Josh Kaplan for insightful discussion, Andrew Chisholm, Zhiping Wang, Anton Maximov, and Fei Chen for critical comments on the manuscript, and our lab members for advice and support. AG is an associate and YJ is an Investigator of the Howard Hughes Medical Institute.

## Additional information

### Funding

| Funder | Grant reference number | Author |
|---|---|---|
| Howard Hughes Medical Institute | | Yishi Jin |
| National Institutes of Health | R01 NS35546 | Yishi Jin |

The funders had no role in study design, data collection and interpretation, or the decision to submit the work for publication.

### Author contributions

KZ, YJ, Conception and design, Acquisition of data, Analysis and interpretation of data, Drafting or revising the article; TMS, Isolated *unc-13(n2609)* allele, Drafting or revising the article; AG, Performed and analyzed the electronic microscopy data

## Additional files

### Supplementary files

• Supplementary file 1. (**A**) Genetic mutations. (**B**) Strains with genetic mutations. (**C**) Transgenes, plasmids, and strains.

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
