## [Decision Letter]

Thank you for sending your work entitled “Position of UNC-13 in the active zone regulates synaptic vesicle release probability and release kinetics” for consideration at *eLife*. Your article has been favorably evaluated by a Senior editor and 3 reviewers, one of whom, Graeme Davis, is a member of our Board of Reviewing Editors.

In the paper by Zhou and colleagues, the authors address how the well-studied protein synaptic protein UNC-13 (Munc13) contributes to synaptic vesicle fusion. Specifically, they evaluate the contribution of a specific domain (C_2_A) of UNC-13 and test the hypothesis that the C_2_A domain helps localize synaptic vesicles near calcium channels at synaptic active zones. The authors show a mild change in the distribution of UNC-13 proteins that either lack the C_2_A domain or have an early stop in the C_2_A domain, and argue that an increased distance from calcium channels, and thus a lower concentration in intracellular calcium, changes release probability. This is a reasonable conclusion given the non-linear dependence of vesicle fusion on intracellular concentration of calcium, but the authors need to more firmly rule out alternative possibilities. Three reviewers agree that this work is suitable for publication in *eLife* and that the recent acceptance of related work will not preclude publication of this work at *eLife* since the two studies were independently conceived and submitted for publication in a similar time frame. The following issues should be addressed:

1) Sucrose responses, as shown, are not a measure of pool size, since there is clearly no major pool depletion. Nevertheless, the fact, that sucrose can elicit a multiple of the eEPSC-charge and that the mutants show similar responses, can be used to argue that priming is intact and that the effect of the mutations is most likely on the Ca-triggering of the release. However, the authors should avoid calling their measured quantities ‘release probabilities’ and ‘pool sizes’. This can be addressed in the text.

2) The interpretation of changes in EPSC-decays as changes in release time course is only valid if these decays are markedly slower than those of spontaneous EPSCs (tEPSCs). Unfortunately the authors analyze only tEPSC frequencies and -amplitudes. An assertion, that tEPSCs are short (in case they are) would be helpful for the reader to understand that slowness is a result of desynchronisation.

3) Decreased protein level of UNC-13 is an alternative explanation for the reduced current observed in the partial loss of function n2609 allele. An early stop makes message sensitive to nonsense-mediated decay or is less efficiently translated. Alternatively, the modified protein might be less stable. An indirect test is to study transgenic worms that overexpress the mutant protein or GFP-tagged variants (or perhaps more easily, the C_2_A-variant). If these fully rescue null mutants, then protein levels are a possible explanation for the partial loss of function in *n2609* mutants.

4) Greater care should be taken in the conclusions when the authors attempt to explain the observed phenotypes based on redistribution of vesicles, which seems very minor.

5) It should be made clear that release is graded in *C. elegans*.

6) The prolonged sucrose stimulation experiments are difficult to evaluate without control experiments showing the extent of release in the *s69* null mutants when using this protocol.

---

## [Author Response]

*1) Sucrose responses, as shown, are not a measure of pool size, since there is clearly no major pool depletion. Nevertheless, the fact, that sucrose can elicit a multiple of the eEPSC-charge and that the mutants show similar responses, can be used to argue that priming is intact and that the effect of the mutations is most likely on the Ca-triggering of the release. However, the authors should avoid calling their measured quantities ‘release probabilities’ and ‘pool sizes’. This can be addressed in the text*.

We apologize for not including the full trace of sucrose response in the original submission. These traces are now shown in the revised Figures 1 and 4. Under our sucrose treatment protocol, we see a complete depletion of the SV pool such that the evoked currents decay to the baseline. We have also provided the recording in *unc-13(s69)* (addressing point 6), which shows little release over the extended sucrose stimulation. However, we agree that this analysis alone is not sufficiently quantitative to measure pool size. In the revised manuscript, we have used “SV priming”, instead of SV “pool size”.

We have revised the conclusion regarding the function of C_2_A domain in SV priming to: “Sucrose-induced charge transfers in the time windows of first one and five seconds were similar between wild type and *unc-13(n2609)* (Figure 1). Both *Si(UNC-13L)* and *Si(UNC-13L*^*C2A-*^*)* transgenes rescued SV priming in *unc-13(s69)* null mutants to the level of wild-type. These results indicate that SVs are fully competent for release in the absence of the C_2_A domain of UNC-13L.”

In the revised manuscript we have included a statement that the ratio of mean charge transfers during eEPSC and under sucrose application “may not directly represent the release probability, but it is positively correlated with release probability”.

*2) The interpretation of changes in EPSC-decays as changes in release time course is only valid if these decays are markedly slower than those of spontaneous EPSCs (tEPSCs). Unfortunately the authors analyze only tEPSC frequencies and -amplitudes. An assertion, that tEPSCs are short (in case they are) would be helpful for the reader to understand that slowness is a result of desynchronisation*.

We appreciate this comment. We have provided 75–25% decay time of tEPSC in a figure supplement for Figure 5. In *C. elegans* NMJs, tEPSCs are short (around 1 ms for 75–25% decay time), whereas the 90–10% decay times of eEPSCs are larger than 10 ms. The decay times of tEPSC among all the recorded genotypes in our study are indistinguishable, indicating that postsynaptic receptor response kinetics is unaltered even for synapses with slower release kinetics.

We have made the recommended assertion (text starting): “In *C. elegans* NMJs, the decays of tonic excitatory postsynaptic current (tEPSC)…”

*3) Decreased protein level of UNC-13 is an alternative explanation for the reduced current observed in the partial loss of function n2609 allele. An early stop makes message sensitive to nonsense-mediated decay or is less efficiently translated. Alternatively, the modified protein might be less stable. An indirect test is to study transgenic worms that overexpress the mutant protein or GFP-tagged variants (or perhaps more easily, the C2A-variant). If these fully rescue null mutants, then protein levels are a possible explanation for the partial loss of function in* n2609 *mutants*.

We have provided such electrophysiological recording data in new figures (Figure 1—figure supplement 4 and Figure 5—figure supplement 1) showing that although both UNC-13L and UNC-13L^C2A-^ overexpression (OE, generated as extrachromosomal arrays using 10ng/μl plasmids of interest) rescued the paralysis of *unc-13(s69)* mutants, UNC-13L^C2A-^ OE did not fully rescue eEPSC amplitude or tEPSC frequency, whereas UNC-13L OE fully rescued synaptic transmission and in fact caused increased tEPSC frequency. These results thus provide further support for our conclusion that the observed synaptic defects of *unc-13(n2609)* is due to specific loss of functional C_2_A domain of UNC-13L.

*4) Greater care should be taken in the conclusions when the authors attempt to explain the observed phenotypes based on redistribution of vesicles, which seems very minor*.

We appreciate this suggestion and have made revision to the conclusion: “the mild reduction in the centrally docked SV in *unc-13(n2609)* may partially account for the reduced release probability”.

*5) It should be made clear that release is graded in* C. elegans.

We have now included such a statement: “SV release at these synapses occurs in a graded manner in response to membrane potential change (40).”

*6) The prolonged sucrose stimulation experiments are difficult to evaluate without control experiments showing the extent of release in the* s69 *null mutants when using this protocol*.

In the revised Figure 1, we have included the data showing that there is little release in *unc-13(s69)* in the first one or five seconds of our prolonged sucrose stimulation protocol. The effect within the first second of sucrose treatment is not different from previous reports using shorter sucrose stimulation (21; 46).